# Opposing roles of nuclear receptor HNF4α isoforms in colitis and colitis-associated colon cancer

**Karthikeyani Chellappa[1]\*†, Poonamjot Deol[1], Jane R Evans[1], Linh M Vuong[1], Gang Chen[2], Nadege Briançon[3], Eugene Bolotin[1], Christian Lytle[2], Meera G Nair[2], Frances M Sladek[1]\***

[1]Department of Cell Biology and Neuroscience, University of California, Riverside, Riverside, United States; [2]Division of Biomedical Sciences, University of California, Riverside, Riverside, United States; [3]Department of Cell Biology, Harvard Medical School, Boston, United States

**\*For correspondence:** kchel@
mail.med.upenn.edu (KC); frances.
sladek@ucr.edu (FMS)

**Present address:** †Institute for Diabetes, Obesity, and Metabolism, Department of Physiology, Perelman School of Medicine, University of Pennsylvania, Philadelphia, United States

**Abstract** HNF4α has been implicated in colitis and colon cancer in humans but the role of the different HNF4α isoforms expressed from the two different promoters (P1 and P2) active in the colon is not clear. Here, we show that P1-HNF4α is expressed primarily in the differentiated compartment of the mouse colonic crypt and P2-HNF4α in the proliferative compartment. Exon swap mice that express only P1- or only P2-HNF4α have different colonic gene expression profiles, interacting proteins, cellular migration, ion transport and epithelial barrier function. The mice also exhibit altered susceptibilities to experimental colitis (DSS) and colitis-associated colon cancer (AOM+DSS). When P2-HNF4α-only mice (which have elevated levels of the cytokine resistin-like β, RELMβ, and are extremely sensitive to DSS) are crossed with *Retnlb⁻/⁻* mice, they are rescued from mortality. Furthermore, P2-HNF4α binds and preferentially activates the RELMβ promoter. In summary, HNF4α isoforms perform non-redundant functions in the colon under conditions of stress, underscoring the importance of tracking them both in colitis and colon cancer.

## Introduction

Hepatocyte nuclear factor 4alpha (HNF4α) (NR2A1) is a highly conserved member of nuclear receptor superfamily of ligand-dependent transcription factors that is expressed in liver, kidney, pancreas, stomach and intestine (*Sladek et al., 1990*). HNF4α is best known for its role in the liver where it is a master regulator of liver-specific gene expression and essential for adult and fetal liver function (*Hayhurst et al., 2001*; *Kaestner, 2010*; *Bolotin et al., 2010*; *Odom, 2004*). HNF4α is also known for its role in the pancreas where it regulates insulin secretion from beta cells (*Gupta et al., 2007*; *2005*; *Miura et al., 2006*). Mutations in the *HNF4A* gene and promoter regions are associated with Maturity Onset Diabetes of the Young 1 (MODY1) (*Ellard and Colclough, 2006*).

In contrast, the role of HNF4α in the intestine has only recently been investigated. Knockout of the *Hnf4a* gene in the embryonic mouse colon results in disrupted crypt topology, and a decreased number of epithelial and mature goblet cells (*Garrison et al., 2006*), while the adult intestinal knockout shows defects in the balance between proliferation and differentiation as well as immune function, ion transport, epithelial barrier function and oxidative stress (*Ahn et al., 2008*; *Cattin et al., 2009*; *Darsigny et al., 2009*; *2010*; *Chahar et al., 2014*). Dysregulation of the *HNF4A* gene is linked to several gastrointestinal disorders including colitis and colon cancer and a single nucleotide polymorphism in the *HNF4A* gene region is associated with ulcerative colitis (*Ahn et al., 2008*; *Chellappa et al., 2012*; *Tanaka et al., 2006*; *Oshima et al., 2007*; *Barrett et al., 2009*).

**eLife digest** The digestive system in animals consists of a network of organs – including the liver, stomach, pancreas and intestines – that work together to break down food and deliver energy to the rest of the body. Many proteins called transcription factors help to guide the development of these organs and keep them healthy throughout life. Among these is a protein called HNF4α. In various diseases of the digestive system, such as gastric cancer or inflammatory bowel disease, the production of HNF4α is not properly regulated.

Gene expression can be activated by transcription factors binding to regions of DNA called promoters. The gene that encodes HNF4α has two promoters called P1 and P2, and each produce several different versions of the HNF4α protein.

The colon contains intestinal glands (also known as colonic crypts) that contain a lower part in which cells actively divide and an upper part of non-dividing cells that help with digestion. Previous studies have shown that if the mouse colon is unable to produce HNF4α, the structure of the crypts is disrupted. By studying crypts taken from the colon of mice, Chellappa et al. have now found that P1-HNF4α proteins are mainly produced at the top of the crypts, whereas P2-HNF4α proteins are found mainly at the bottom.

Chellappa et al. then used two sets of genetically engineered mice: one that can only produce P1-HNFα proteins, and one that only has P2-HNFα proteins. Under normal conditions both sets of mice appeared healthy. However, differences became apparent if the mice were subjected to treatments that cause colitis or colitis-associated colon cancer. Mice that could only produce P1-HNF4α proteins were less susceptible to colitis and got fewer and smaller tumors than normal mice. By contrast, mice that could only produce P2-HNF4α experienced more colitis and developed more tumors than normal mice.

Comparing the genes expressed in the colon cells of these two types of mice revealed several differences. In particular, much more of a pro-inflammatory protein called RELMβ was produced in P2-only mice. Chellappa et al. then proceeded to show that RELMβ is essential for the susceptibility of P2-mice to coliltis.

Overall, the experiments show that P1-HNF4α and P2-HNF4α perform different tasks both in the healthy and the diseased mouse colon. In future it will be important to work out how the balance between the two sets of proteins is disrupted in diseases of the colon.

While it is clear that HNF4α is critical for normal colon function, it is not known which transcript variant is the most relevant. There are two different promoters (proximal P1 and distal P2) in the HNF4α gene that are both active in the colon. The promoters are conserved from frog to human and, along with alternative splicing, give rise to nine different transcript variants of HNF4α (*Huang et al., 2009*) (*Figure 1A*). The major isoforms of the P1 promoter are HNF4α1/α2 while the P2 promoter gives rise to HNF4α7/α8: distinct first exons result in an altered A/B domain which harbors the activation function 1 (AF-1) while the DNA and ligand binding domains are identical. The two promoters are expressed under unique temporal and spatial conditions, with the large and small intestine being the only adult tissues that express both P1- and P2-HNF4α (*Tanaka et al., 2006*; *Nakhei et al., 1998*). While a loss of P1- but not P2-HNF4α has been noted in colon cancer (*Chellappa et al., 2012*; *Tanaka et al., 2006*), the specific roles of the HNF4α isoforms remain obscure. For example, P1-driven HNF4α acts as a tumor suppressor in mouse liver (*Hatziapostolou et al., 2011*; *Walesky et al., 2013a*). In contrast, the *HNF4A* gene and protein are amplified in human colon cancer (*Cancer Genome Atlas Network, 2012*; *Zhang et al., 2014*) although the different isoforms were not distinguished in those studies. We recently showed that ectopic expression of P1- but not P2-HNF4α decreased the tumorigenic potential of the human colon cancer cell line HCT116 in a mouse xenograft model (*Vuong et al., 2015*), suggesting that the different HNF4α isoforms may indeed play distinct roles in the colon.

Here, we investigate the role of P1- and P2-HNF4α isoforms in the mouse colon using genetically engineered mice that express either the P1- or the P2-HNF4α isoforms (*Briançon and Weiss, 2006*). We show that in wildtype (WT) mice P1- and P2-HNF4α are expressed in different compartments in

the colonic epithelium, interact with distinct sets of proteins, regulate the expression of unique sets of target genes, and play distinct roles during pathological conditions such as colitis and colitis-associated colon cancer (CAC). We also provide genetic and biochemical evidence indicating that RELMβ, a member of the RELM/FIZZ family of cytokines, plays a critical role in the response of HNF4α to colitis and appears to be both directly and indirectly regulated by HNF4α.

## Results

### Compartmentalization of P1- and P2-HNF4α in mouse colonic epithelium

In the distal colon, the bottom two-thirds of the crypt and the top one-third, including surface epithelium, are functionally categorized as proliferative and differentiated compartments, respectively (*Potten et al., 1997*). We used monoclonal antibodies specific to the different HNF4α isoforms (*Chellappa et al., 2012*; *Tanaka et al., 2006*) (*Figure 1A*) to examine the distribution of P1- and P2-HNF4α along the crypt-surface axis. The P1/P2 antibody, which recognizes both P1- and P2-HNF4α, shows HNF4α expression in both crypt and surface epithelial cells (*Figure 1B*), as reported previously (*Ahn et al., 2008*; *Darsigny et al., 2009*; *Chahar et al., 2014*). In contrast, the isoform-specific antibodies reveal that P1-HNF4α is expressed mainly in the differentiated compartment, not in the proliferative compartment as defined by NKCC1 staining (*Figure 1C*). P2-HNF4α was observed primarily in the bottom half of the crypt (*Figure 1B*) and co-localized with the proliferation marker Ki67 in isolated colonic crypts (*Figure 1D*). While there was some expression of P2-HNF4α in the differentiated compartment (i.e., non Ki67 expressing cells), it was notably absent from the surface epithelium (*Figure 1B*).

### Isoform-specific dysregulation of HNF4α in mouse models of colitis and colon cancer

Previous studies showed that *HNF4A* expression is decreased in human inflammatory bowel disease (IBD) patients and intestine-specific deletion of the mouse *Hnf4a* gene increases susceptibility to dextran sodium sulfate (DSS)-induced colitis (*Ahn et al., 2008*) and can lead to chronic inflammation even in the absence of DSS (*Darsigny et al., 2009*). However, these studies do not address the role of the individual HNF4α isoforms. We treated young adult male mice (WT) with 2.5% DSS and found a statistically significant decrease in total HNF4α following 5 days of DSS treatment, as others have observed (*Ahn et al., 2008*; *Chahar et al., 2014*), and an increase in HNF4α during the recovery phase, especially P1-HNF4α (*Figure 2A,B*). Contrary to the restricted expression of P1-HNF4α in the differentiated compartment in untreated mice, P1-HNF4α was also expressed near the bottom of the crypt after DSS treatment (*Figure 2C*), consistent with substantial loss of proliferating cells following DSS treatment (*Tessner et al., 1998*).

In a mouse model of colitis-associated colon cancer (CAC) in which a single injection of azoxymethane (AOM) is followed by multiple treatments of DSS in the drinking water, we found that P1-HNF4α is greatly reduced in tumors compared to untreated controls but that total HNF4α protein was only marginally reduced (*Figure 2D*), suggesting that P2-HNF4α was not affected. The P1-HNF4α decrease correlated with an increase in active Src (pSrc), consistent with our earlier finding that Src specifically phosphorylates and causes the degradation of human P1- but not P2-HNF4α (*Chellappa et al., 2012*). We also observed a specific loss of P1-HNF4α protein in a mouse model of sporadic, non-colitis colon cancer (*Figure 2E*), as we observed previously in humans (*Chellappa et al., 2012*).

### Differential susceptibility of HNF4α isoform-specific mice to colitis-associated colon cancer

To decipher the function of the HNF4α isoforms in the colon, we utilized HNF4α isoform-specific mice generated by an exon swap strategy (*Figure 3A* top left) (*Briançon and Weiss, 2006*). These mice express exclusively either P1-HNF4α (α1HMZ) or P2-HNF4α (α7HMZ) wherever HNF4α is endogenously expressed. Immunoblot analysis confirmed that the HNF4α protein level in the distal colon of the exon swap mice is equivalent to that of WT littermates, and that P2-HNF4α is the major isoform in the distal colon (*Figure 3A* top right and *Figure 3—figure supplement 1E*). In α1HMZ

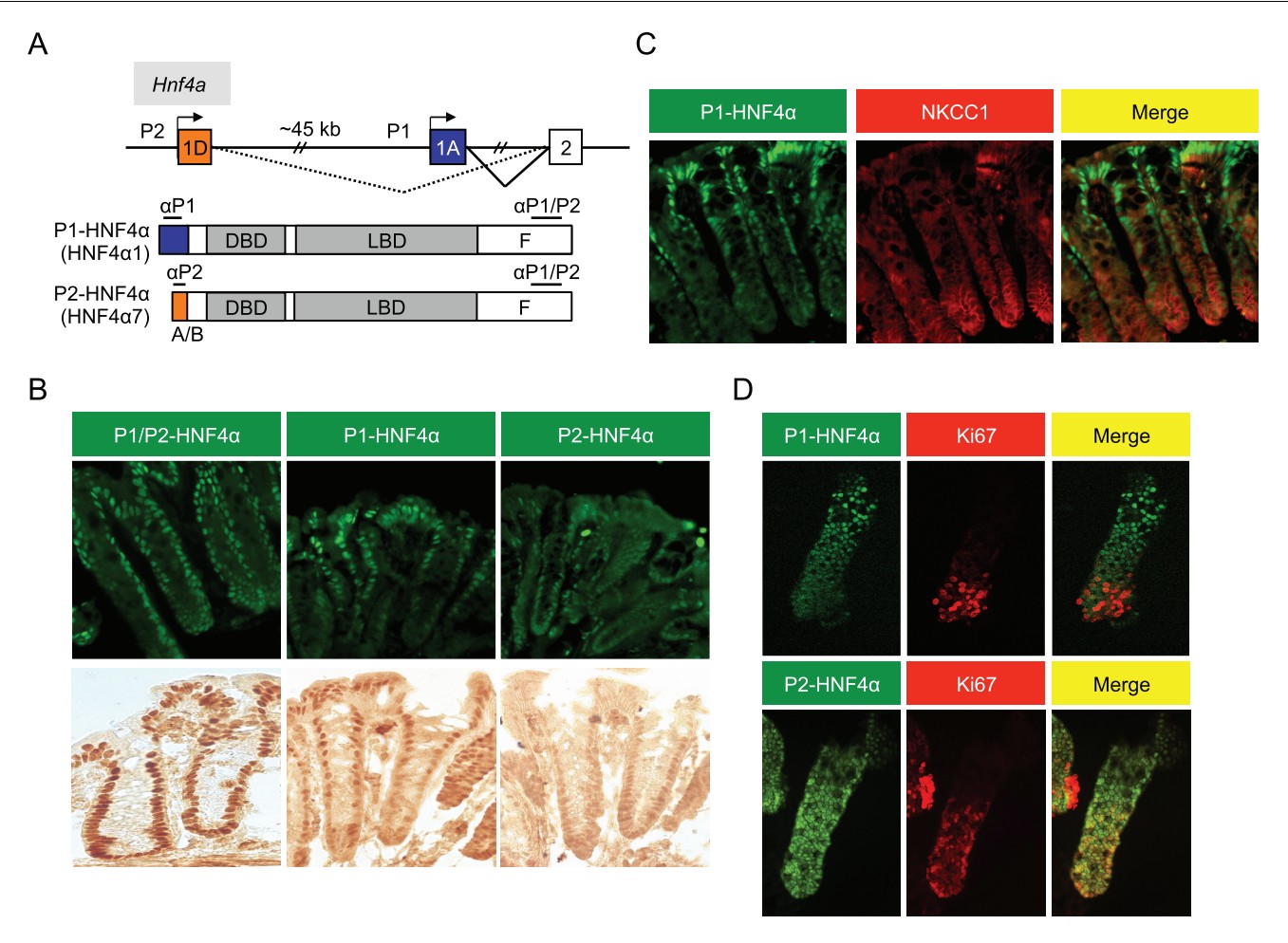

**Figure 1.** Differential localization of HNF4α isoforms in mouse colonic crypts. (**A**) Schematic of the mouse (and human) *Hnf4a* gene showing the two promoters (P1 and P2) (top) and the P1- and P2-driven HNF4α isoforms that they express (bottom). The differential N-terminal A/B domain (indicated in blue and orange) as well as epitopes to isoform-specific (αP1 and αP2) and common (αP1/P2) antibodies are indicated. DBD, DNA binding domain; LBD, ligand binding domain; F, F domain. (**B–D**) IF and immunohistochemistry of distal colon (**B,C**) or isolated colonic crypts (**D**) stained for the indicated proteins using the antibodies in (**A**) (**B**: 40X magnification; **C,D**: 25X magnification with digital zoom). NKCC1 (*Slc12a2*) (**C**) and Ki67 (**D**) mark the proliferative compartment of the crypt. Representative images from two independent experiments (n=2–4 mice per genotype) are shown.

mice, P1-HNF4α was detected in all epithelial cells in both the bottom of the crypt and the surface epithelium; a similar ubiquitous expression was observed for P2-HNF4α in α7HMZ mice (*Figure 3A* bottom).

After 53 days of AOM+DSS treatment, the α1HMZ mice had significantly fewer and smaller tumors compared to WT controls (*Figure 3B*). In addition, despite similar crypt length in untreated WT and α1HMZ mice, the α1HMZ mice did not exhibit the characteristic increase in crypt length associated with mutagen exposure observed in WT mice (*Richards, 1977*) (*Figure 3C*). In fact, the crypt length decreased compared to both treated WT and untreated α1HMZ. We also observed fewer infiltrating immune cells (*Figure 3C* right) as well as decreased spleen-to-body weight ratio in the treated α1HMZ mice (*Figure 3—figure supplement 1A*). After 85 days of treatment, the difference in tumor number was less pronounced: a significant decrease in tumor number was observed in α1HMZ mice only in the smallest tumors (0–2 mm) (*Figure 3D*).

In contrast to α1HMZ mice, the α7HMZ mice exhibited a greater tumor load and tumor number than their WT controls after 53–64 days of treatment (*Figure 3E*). However, at the later time point (85 days), the effect was lost mainly due to increased tumor burden in the WT mice (*Figure 3F* and

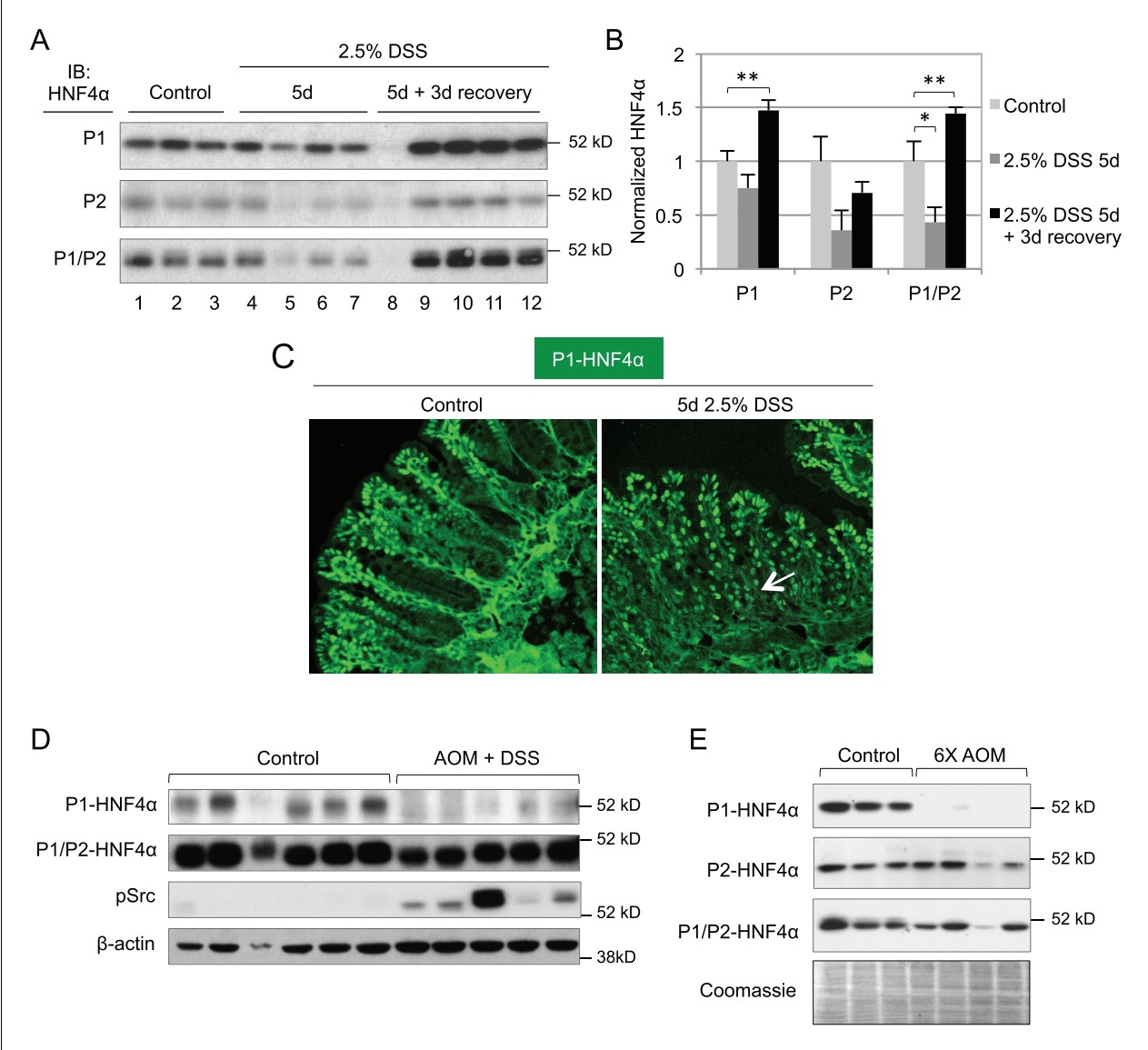

**Figure 2.** Dysregulation of P1- and P2-HNF4α in mouse models of colitis and colon cancer. (**A**) IB of WCE from the distal colon of WT mice treated with 2.5% DSS for 5 days followed by 0 or 3 days recovery, and an analogous region of untreated (Control) mice. Each lane is from a different mouse. The position of the molecular weight marker (52 kD) is shown. (**B**) Quantification of the HNF4α signal in (**A**) normalized to total protein, as determined by Coomassie staining of the same blot. For the purposes of quantification the outlier in lane 8 was omitted. *$P<0.05$, **$P<0.005$. (**C**) Representative IF of distal colon from untreated and DSS-treated WT mice (n=3–4 per condition) stained with P1-HNF4α antibody (40X magnification). Arrow indicates P1-HNF4α expressing cells near the bottom of the crypt in the DSS-treated mice. (**D**) IB as in (**A**) but from the tumor area of WT mice treated with 10 mg/kg AOM and three cycles of a 7-day DSS treatment and harvested at ∼95 days. Three gels were run in parallel with the same extracts; one representative β-actin stain is shown. (**E**) IB analysis as in (**D**) but from mice injected six times with 10 mg/kg AOM and harvested at ∼150 days. Shown is one representative of the three Coomassie stains performed for loading verification.

*Figure 3—figure supplement 1B*). Interestingly, there was no difference in the percent of Ki67-staining cells between WT and α7HMZ mice (53–64 day treatment) (*Figure 3—figure supplement 1C,D*).

## Differential susceptibility of HNF4α isoform-specific mice to colitis

More striking than tumor induction by AOM+DSS in the α1HMZ and α7HMZ mice was their response to an acute DSS treatment to induce colitis –2.5% DSS in drinking water for 5 days. There was a ∼73% mortality rate for α7HMZ mice that occurred starting after three days of recovery when the mice were switched to normal tap water (*Figure 4A*). During the recovery phase, α7HMZ mice

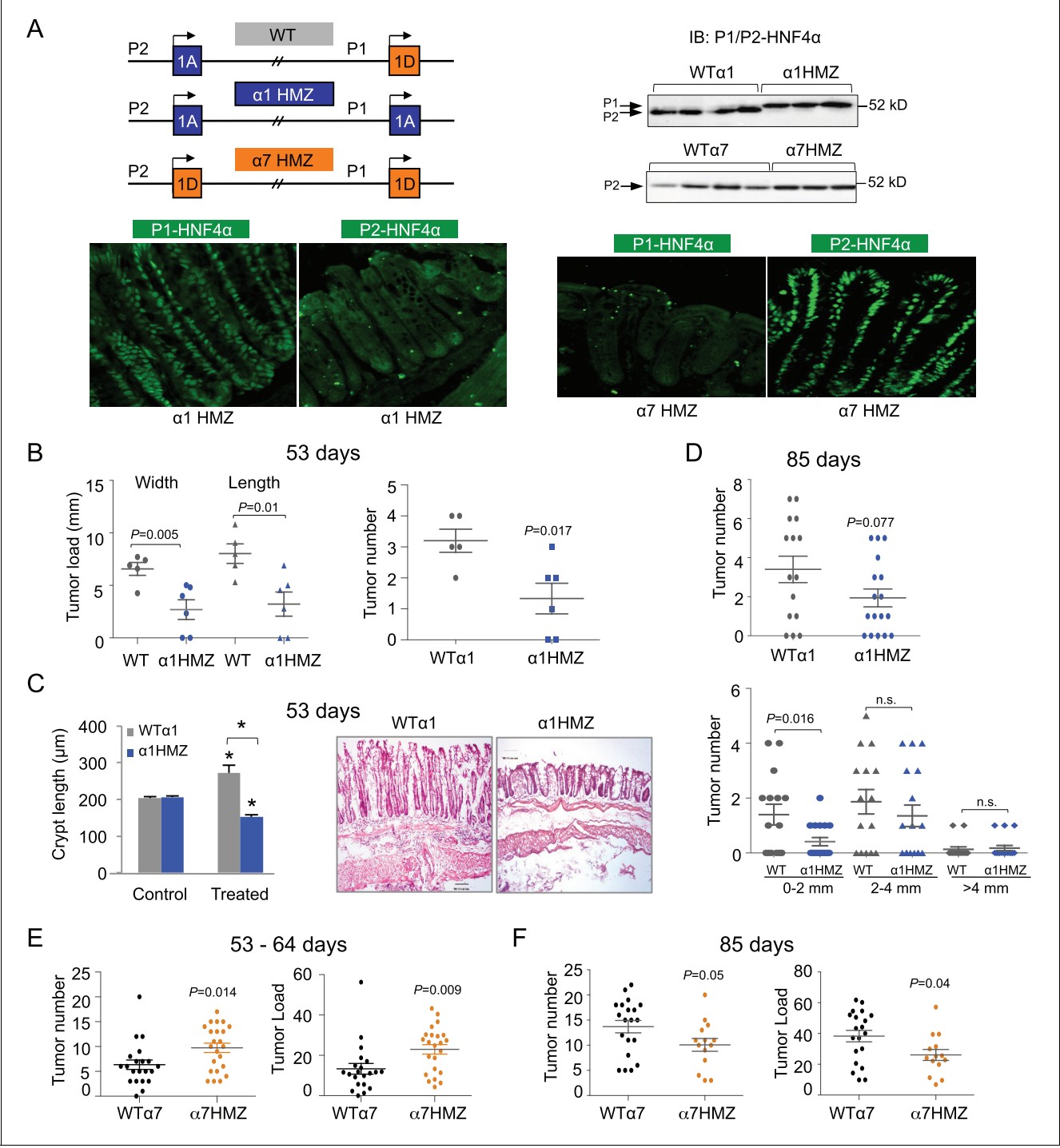

**Figure 3.** Differential susceptibility of HNF4α isoform-specific mice to colitis-associated colon cancer. (**A**) Top left, Schematic of *Hnf4a* exon-swap (i.e., isoform-specific) mice. Top right, IB as in *Figure 2A* of WCE from the distal colon of the exon-swap mice and their WT controls, probed with the common αP1/P2 antibody. See *Figure 3—figure supplement 1E* for verification of protein loading. Bottom, representative IF of distal colons from untreated α1HMZ and α7HMZ mice stained with either P1- or P2-driven HNF4α specific antibodies (40X magnification). N=3–4 mice per genotype examined. Scale for P1-HNF4α α1HMZ is 0.22 x 0.22 microns; all others are 0.36 x 0.36 microns. (**B**) Tumor growth in WTα1 (n = 5) and α1HMZ (n = 6) mice treated with 10 mg/kg AOM and two cycles of DSS (5 days per cycle) and harvested at ~53 days. Right, number of tumors per mouse colon. Left, tumor load (sum of the width or length of all macroscopic lesions in a given mouse). Each symbol represents results from one mouse. (**C**) Left, average length of crypt in WTα1 and α1HMZ mice, untreated (Control) or treated as in (**B**) N = 2–3 mice per condition; 26–56 crypts per mouse were measured. *P<0.0005 between treated and control within a genotype and across genotypes in the treated condition. Right, Representative H&E stain (10X magnification) of mice treated as in (**B**) Scale bar is 100 microns. (**D**) Tumor number in WTα1 (n = 15) and α1HMZ (n = 17) male mice treated as in (**B**)

*Figure 3 continued on next page*

*Figure 3 continued*

but with three cycles of DSS (two cycles of 5 days and one cycle of 4-days) and harvested at ~85 days. Top, total number of tumors per mouse. Bottom, number of tumors per mouse based on the tumor width. n.s., non-significant. (**E**) As in (**B**) but for WTα7 (n = 21) and α7HMZ (n = 23) mice treated with 10 mg/kg AOM and 2–3 cycles of DSS (4–5 days per cycle) and harvested at ~53–64 days. *P*-values between α7HMZ and WTα7 mice are indicated. Tumor data were pooled from three independent experiments. (**F**) Tumor number and load in WTα7 (n = 20) and α7HMZ (n = 14) mice as in (**E**) but harvested at ~85 days. The following figure supplement is available for *Figure 3*:

The following figure supplement is available for figure 3:

**Figure supplement 1.** HNF4α isoform-specific mice subjected to AOM+DSS to induce colitis-associated colon cancer.

exhibited a significant decrease in body weight and colon length (*Figure 4B* and *Figure 4—figure supplement 1A*), and a worse histological score (due to more severe crypt damage, inflammation and ulceration) compared to their WT littermates (*Figure 4C* and *Figure 4—figure supplement 1B*). There was also an increased spleen-to-body weight ratio (*Figure 4—figure supplement 1C*) when the mice were maintained and treated in an open access vivarium. IB analysis revealed that, in contrast to the WT mice that lost expression of both HNF4α isoforms after five days of DSS and then had an increase in P1-HNF4α expression at 3-day recovery (*Figure 2A*), in the α7HMZ mice P2-HNF4α protein amount is notably increased upon DSS treatment and then decreased after a 3-day recovery, as observed by both IB and IF (*Figure 4D*). At 12 days of recovery, we observed a massive infiltration of immune cells and a continued striking loss in crypt structure in α7HMZ mice compared to WT mice (*Figure 4—figure supplement 1D*).

In contrast to the extreme sensitivity of the α7HMZ mice to DSS-induced colitis, α1HMZ mice were less susceptible than their WT controls as indicated by increased colon length (*Figure 4E*) and well-preserved crypt structure and decreased histological score (*Figure 4F*). There was no difference in spleen-to-body weight ratio between the α1HMZ mice and WT controls (data not shown).

Clinical and histological changes occurring a few weeks after DSS treatment are referred to as chronic or advanced changes (*Perše and Cerar, 2012*). To examine chronic effects, we allowed the mice to recover for 18 days after a somewhat milder DSS treatment (4 days) to reduce the mortality of α7HMZ mice. Despite the shorter DSS treatment, after 18 days, the α7HMZ mice still exhibited elevated spleen-to-body weight ratio, increased crypt length, more visibly inflamed colons and immune cell infiltration and overall higher histological scores compared to α1HMZ mice (*Figure 4—figure supplement 2A–D*).

## Transcriptomic and proteomic profile of colons from HNF4α isoform-specific mice

Expression profiling of the distal colon revealed a significant change in a substantial number of genes in the untreated isoform-specific mice compared to their WT controls (*Figure 5—figure supplement 1A*). There was an overall greater effect in α1HMZ than α7HMZ mice in terms of the number of dysregulated genes with a large fold change, which could be due to the fact that P1-HNF4α typically has a more potent transactivation function than P2-HNF4α (*Eeckhoute et al., 2003*). On the other hand the number of genes altered at lower fold change was higher in α7HMZ compared to α1HMZ mice, consistent with more P2-HNF4α protein in the distal colon of WT mice than P1-HNF4α (*Figure 3A*) . Gene Ontology (GO) analysis of the differentially regulated genes showed that in α1HMZ mice there is a marked upregulation of genes involved in wound healing and immune response, as well as a variety of metabolic processes typically associated with differentiation (*Figure 5A*). In contrast, in α7HMZ mice there is a significant upregulation of genes involved in cell cycle and DNA repair and a decrease in genes involved in cell adhesion, motility and ion transport (*Figure 5B*). (See *Figure 5—source data 1A*-1G for the fold change in the top 100 dysregulated genes and the genes in the aforementioned GO categories, respectively).

The DNA binding domains of P1- and P2-HNF4α are 100% identical and the isoforms have similar in vitro DNA binding specificity and chromatin immunoprecipitation (ChIP)-seq profiles in human colon cancer cells (*Vuong et al., 2015*). Therefore, to elucidate the mechanism responsible for differential gene expression in mouse colon, we performed RIME (Rapid Immunoprecipitation Mass spectrometry of Endogenous proteins) on HNF4α in the colons of α1HMZ and α7HMZ mice (*Figure 5C*).

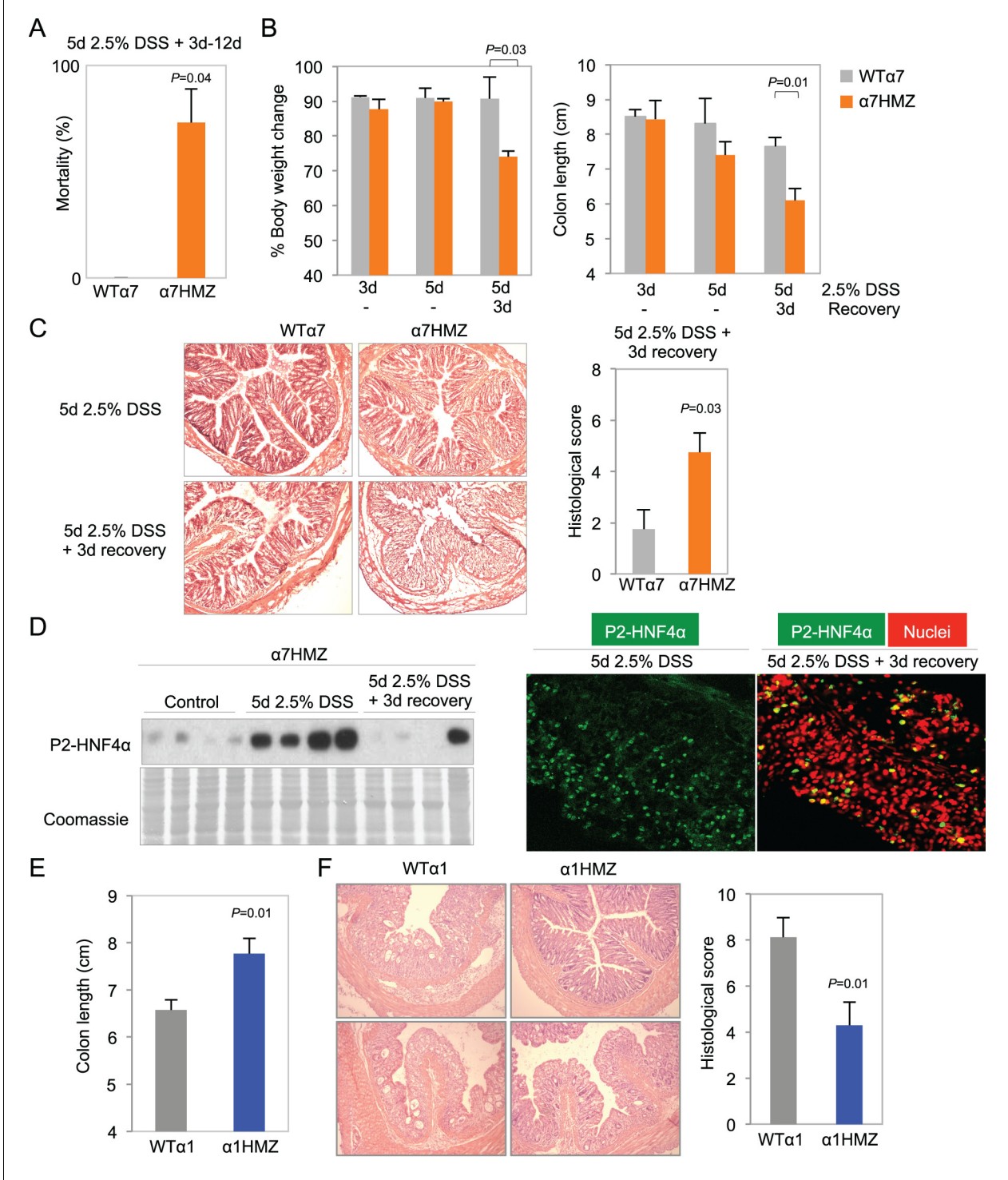

**Figure 4.** Differential susceptibility of HNF4α isoform-specific mice to DSS-induced colitis. (**A**) Percent mortality of WTα7 (n = 28) and α7HMZ (n = 16) mice treated with 2.5% DSS for 5 days. α7HMZ mice typically died during day 3 to 12 of recovery following DSS treatment. Data pooled from two independent experiments. Not shown is a third experiment with older mice (21–23 weeks) with similar results (WT: 1 of 5 mice died; α7HMZ: 3 of 6 mice died). (**B**) Change in bodyweight (represented as% initial body weight) (left) and colon length (right) of WT (n = 4) and α7HMZ (n = 4) mice treated as indicated. Significant comparisons are indicated with a *P*-value. (**C**) Left, representative H&E stain of WT and α7HMZ mice treated with 2.5% DSS for 5 days followed by 0 or 3 days of recovery. Right, histological scores of colitis in WTα7 (n = 4) and α7HMZ (n = 4) mice. (**D**) Left, IB for HNF4α (P1/P2 antibody) of WCE from the distal colon of α7HMZ mice treated as indicated. Right, representative IF of distal colon from α7HMZ mice treated with 2.5% DSS for 5 days -/+ recovery as indicated and stained with P1/P2-HNF4α antibody (green) and TO-PRO3 (red) for nuclei (40X magnification).
*Figure 4 continued on next page*

Figure 4 continued

Extracts from four mice per genotype (out of n = 5–7) were randomly chosen for IB analysis on a single gel/blot; sections from 3 mice per genotype were examined. (**E**) Colon length of WT (n = 8) and α1HMZ (n = 10) male mice treated with 2.5% DSS for 5 days followed by 3 days of recovery. Results from two independent experiments were pooled. (**F**) Representative H&E stain (left) and histological scores (right) of colitis in WTα1 (n = 8) and α1HMZ (n = 10) mice treated as in (**E**). The following figure supplements are available for *Figure 4:*

The following figure supplements are available for figure 4:

**Figure supplement 1.** Increased susceptibility of α7HMZ mice to DSS-induced colitis.

**Figure supplement 2.** Increased inflammation in α7HMZ mice in DSS-induced colitis.

The isoforms share 76 interacting proteins, including previously reported HNF4γ (*Daigo et al., 2011*), a well known co-regulator for nuclear receptors (NRIP1, RIP140) and DPF2, a BRG1-associated factor (BAF45). However, there were more proteins uniquely binding HNF4α in α7HMZ and α1HMZ colons – 138 and 99, respectively (*Figure 5C* top and *Figure 5C—source data 2A–E*). Src tyrosine kinase, for example, bound uniquely in α1HMZ colons, consistent with our previous report that Src preferentially phosphorylates and interacts with HNF4α1 in cell-based and in vitro systems (*Chellappa et al., 2012*) and validating RIME for identification of differential interacting proteins in vivo. In contrast, CUL4A, a core component of a cullin-based E3 ubiquitin ligase complex and overexpressed in cancer (*Kopanja et al., 2009*), and PCM1, a centrosome binding protein translocated to the JAK2 locus in certain leukemias (*Reiter et al., 2005*), both bound uniquely in α7HMZ colons. Both CUL4A and PCM1 are required for efficient cell proliferation, genome stability and/or proper centrosome function (*Erger and Casale, 1998*; *Farina et al., 2016*), consistent with the upregulation of genes involved in cell cycle and DNA repair in α7HMZ colons (*Figure 5B*), and accelerated tumorigenesis in α7HMZ mice (*Figure 3E*).

Cross-referencing the interacting proteins to those in the literature associated with colon cancer and inflammatory bowel disease (IBD) revealed several additional relevant proteins for each genotype, the vast majority of which (62.9%) are known transcription regulators, protein kinases or phosphatases and associated proteins (*Figure 5C*). For example, NDRG2, a kinase downstream of the mTOR/SGK pathway and a tumor suppressor that mediates apoptosis (*Deuschle et al., 2012*), and EMD, a nuclear membrane protein phosphorylated by Src (*Tifft et al., 2009*), both preferentially bind HNF4α1 and have been negatively associated with colon cancer. Likewise, HNF4α1 was preferentially bound by catalytic subunits of AMPK (PRKAA1/2) and is known to be phosphorylated by AMPK, which decreases its protein stability (*Hong et al., 2003*). However, AMPK suppresses cell proliferation via inhibition of mTOR and activation of p53 pathways (*Motoshima et al., 2006*) and low levels of AMPK activity are correlated with poor survival in metastatic colon cancer patients (*Zulato et al., 2014*), indicating that additional studies are required to elucidate the impact of AMPK signaling on HNF4α in colitis and colon cancer. In contrast, protein kinase C beta 2 (PRKCB) preferentially interacts with HNF4α7 and is known to be both necessary and sufficient to confer susceptibility to AOM-induced colon carcinogenesis in the colonic epithelium (*Liu, 2004*). All told, there were hundreds of proteins that preferentially interacted with the HNF4α isoforms, including many signaling molecules as well as RNA binding proteins and transcription factors, providing multiple potential mechanisms for differential gene expression.

## Differential effects on cell migration and chloride secretion in HNF4α isoform-specific mice

While the isoform-specific mice did not exhibit any overt morphological abnormalities in their intestines under normal conditions, the gene expression analysis (and AOM/DSS and DSS colitis results) suggested potential functional differences. Since there was a decrease in cell motility genes in α7HMZ mice, we examined migration of BrdU-labeled cells up the crypt and found that 48 hr after injection the average position of the BrdU-labeled cells (both in absolute terms and relative to the bottom of the crypt) was lower in α7HMZ mice compared to WT: this resulted in a statistically significant decreased migration of the BrdU+ cells during the 45 hr period (*Figure 5D* and *Figure 5—figure supplement 1B,C*). Conversely, the position of the BrdU+ cells, and hence migration, was

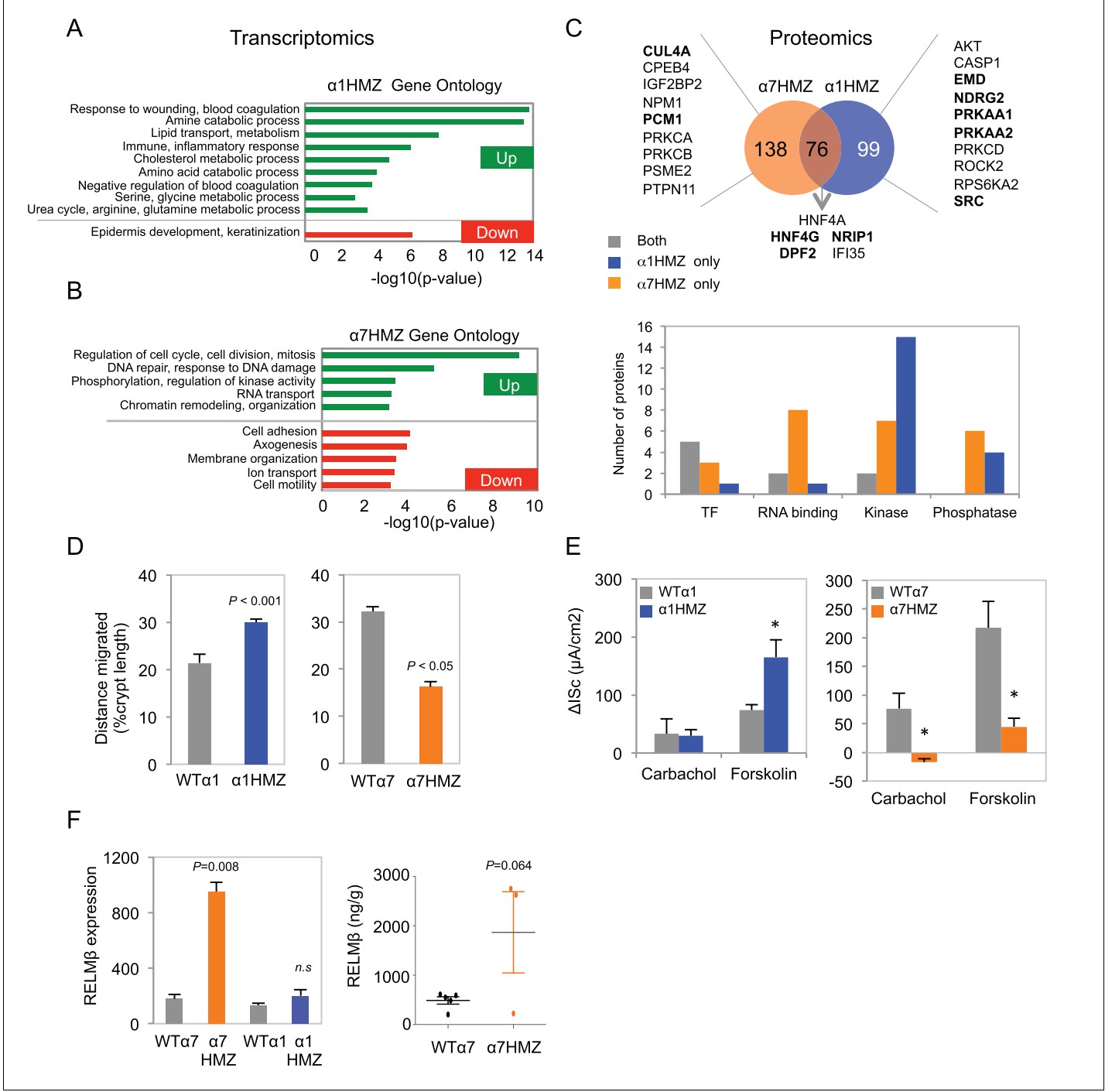

**Figure 5.** Altered gene expression, interacting proteins, migration and ion transport in HNF4α isoform-specific mice. (A,B) Comparative Gene Ontology (GO) of genes differentially regulated (≥two-fold) in the distal colon of untreated α1HMZ (A) and α7HMZ (B) mice. (C) Top, Venn diagram of total number of HNF4α-interacting proteins from RIME analysis found in α7HMZ only, α1HMZ only or both α7HMZ and α1HMZ colons, as described in Material and methods. Indicated are nuclear proteins that have been implicated in regulating gene expression and associated with human or mouse colon cancer, IBD, Crohn's disease and/or ulcerative colitis, as well as other pro-proliferative proteins found only in α7HMZ colons. Shown also are transcription factors that interact with HNF4α in both genotypes. Bold, proteins mentioned in text. Bottom, Total number of proteins in the indicated categories that show a significant interaction with HNF4α in the exon swap mice. TF, transcription factor; RNA binding proteins; kinase and phosphatase categories include only protein kinases and phosphatases, as well as relevant scaffolding proteins b. (D) Untreated HNF4α isoform-specific mice and their WT littermates (n = 3–4 per genotype) were injected with BrdU (75 mg/kg) and sacrificed at 2 hr or 48–50 hr. The distance migrated by the BrdU$^+$ cells from the bottom of the crypt between 2 hr and 48–50 hr is plotted as% crypt length; 5–38 crypts per mouse were scored. (E) Intestinal

*Figure 5 continued on next page*

*Figure 5 continued*

chloride secretion in response to 10 µM forskolin and 100 µM carbachol represented as change in short-circuit current (ΔIsc). Left, WTα1 (n = 6) and α1HMZ (n = 5–8) mice. *P<0.02 between α1HMZ and WTα1. Right, WTα7 (n = 4) and α7HMZ (n = 3) mice. *P<0.05 versus WTα7. Results from one experiment per genotype are shown: a second independent experiment for α7HMZ yielded similar results (not shown). (F) Left, RELMβ mRNA expression in the distal colon of untreated α1HMZ, α7HMZ and their WT controls from microarrays in (A,B), represented as an average of the three *Retnlb* probes. P<0.008 versus WTα7. Right, RELMβ protein level quantified by ELISA in the mid colon homogenate of untreated WTα7 (n = 5) and α7HMZ (n = 3) mice. Shown are means of technical triplicates. The following supplementary figure and source data are available for *Figure 5*:

The following source data and figure supplement are available for figure 5:

**Source data 1.** Transcriptomic analysis of HNF4α isoform-specific mice.
**Source data 2.** Proteomic analysis of HNF4α isoform-specific mice.
**Figure supplement 1.** Transcriptomic and BrdU analysis of HNF4α isoform-specific mice.

significantly higher in α1HMZ mice (*Figure 5D* and *Figure 5—figure supplement 1D,E*). Despite these differences, there was a similar number of total BrdU+ cells in WT and α7HMZ mice (*Figure 5—figure supplement 1F*).

Since the ion transport genes were also decreased in α7HMZ, we examined electrogenic chloride secretion in isolated colonic mucosa. The distal colon of WT and α7HMZ mice exhibited a similar transmucosal electrical resistance and basal Isc (data not shown). However, the α7HMZ distal colon was refractory to stimulation with calcium-dependent (carbachol) and cAMP-dependent chloride secretagogues (forskolin), while the α1HMZ distal colon showed a markedly increased Isc response to forskolin (*Figure 5E*). Since impaired chloride secretion is observed in colitis (*Hirota and McKay, 2009*), this differential response to forskolin as well as cell migration could explain, at least in part, the differential sensitivity of the α1HMZ and α7HMZ mice to DSS.

## Elevated RELMβ expression in α7HMZ mice plays a role in DSS sensitivity

During experimental colitis the cytokine RELMβ is known to activate the innate immune system in response to loss of epithelial barrier function and increased exposure to gut microbiota: hence, mice lacking the *Retnlb* gene are known to be resistant to experimental colitis (*Hogan et al., 2006*; *McVay et al., 2006*). Interestingly, one of the genes most significantly upregulated in the untreated α7HMZ colon was *Retnlb* (5.3-fold increase versus WT controls); RELMβ protein levels were also increased (*Figure 5F*). In contrast, there was no significant difference in RELMβ gene expression between α1HMZ mice and their WT littermates (*Figure 5F* left).

To determine whether RELMβ plays a causal role in the susceptibility of α7HMZ mice to colitis, we crossed α7HMZ mice with a RELMβ knock out (*Retnlb*$^{-/-}$) to generate RbKO/α7HMZ mice (*Figure 6—figure supplement 1A*). We confirmed that RELMβ expression is lost in these mice, that HNF4α protein levels are unchanged by the RELMβ knock out and that the α7HMZ allele has the same effect in the 'Rb line' (designated C57BL/6N+J, see Materials and methods for details) in terms of body weight loss and increased RELMβ expression after DSS treatment (*Figure 6A* and *Figure 6—figure supplement 1B–D*).

Interestingly, while the histological score of the RbKO/α7HMZ mice was somewhat improved compared to WT/α7HMZ at three days of recovery, the difference was not significant (P=0.13) (*Figure 6B*). In contrast, the weight loss of the RbKO/α7HMZ mice was completely restored to WT/WT levels (*Figure 6C*), as was the colon length (*Figure 6D*) and overall survival (*Figure 6E*). Notably, the RELMβ protein levels in the α1HMZ mice were significantly reduced at three days of recovery (although elevated right at the end of the DSS treatment) and inversely correlated with colon length (*Figure 6F*). These results suggest that elevated RELMβ expression in untreated and DSS-treated α7HMZ mice and decreased expression during recovery in α1HMZ mice might contribute to their increased and decreased susceptibility to DSS-induced colitis, respectively. Interestingly, body weight loss is attenuated in α1HMZ during DSS treatment, however this protective effect is lost during recovery (*Figure 6—figure supplement 1E*). All together our results suggest that both P1- and P2-HNF4α are critical for full recovery following DSS treatment.

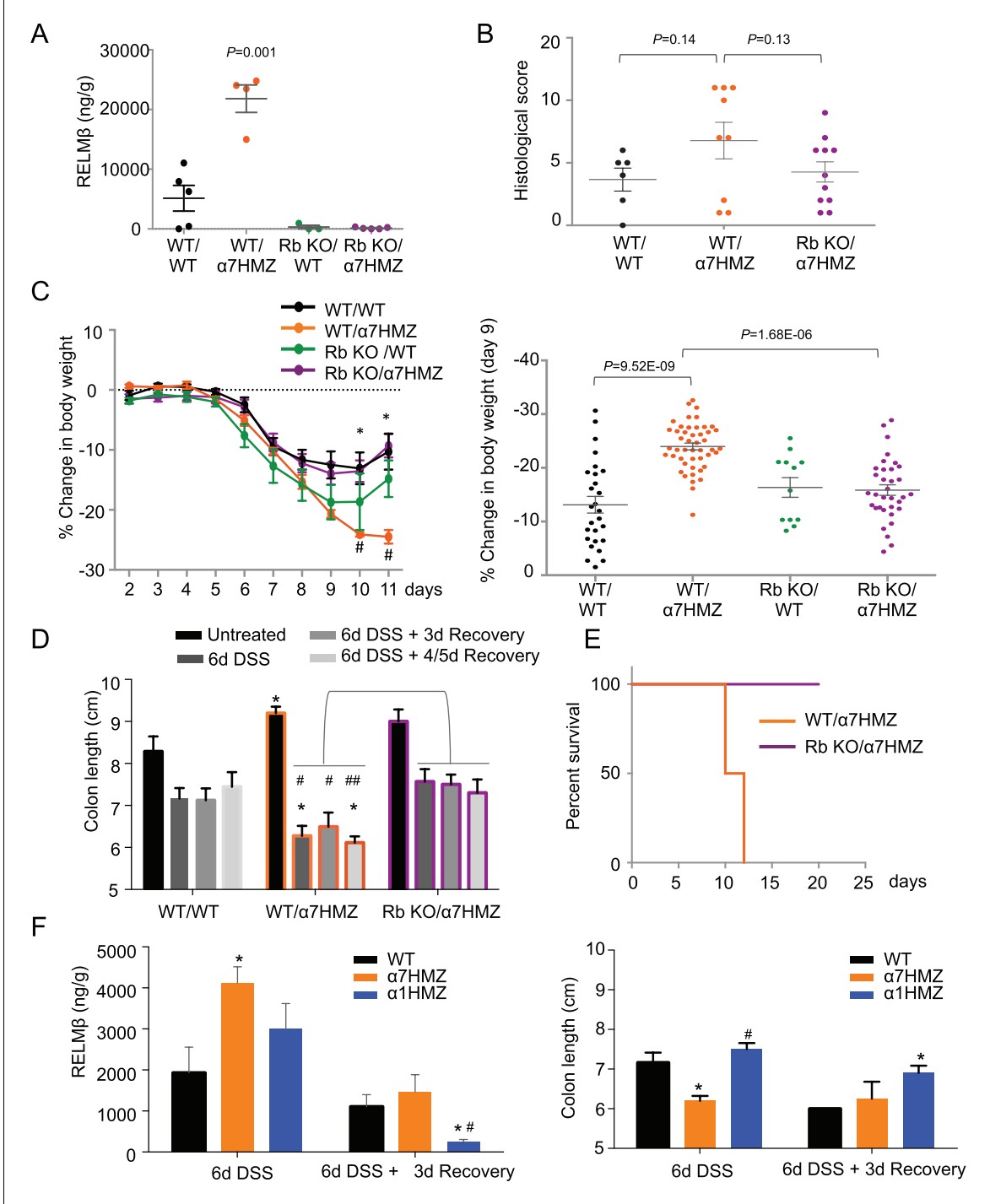

**Figure 6.** RELMβ knockout decreases susceptibility of α7HMZ mice to colitis. (**A**) RELMβ protein level quantified by ELISA in the midcolon homogenate of mice with the indicated genotype treated with 2.5% DSS for 6 days. Genotypes are indicated as *Retnlb/Hnf4a*. N = 3–5 mice per genotype as indicated by each dot. (**B**) Histological scores of colitis in WT/WT (n = 6), WT/α7HMZ (n = 9) and Rb KO/ α7HMZ (n = 11) male mice treated with 2.5% DSS for 5 days followed by 3 days of recovery. Multiple sections per mouse were scored. (**C**) Percent change in body weight during and following DSS treatment (2.5% for 6 days). Day 0 is the start of treatment. Left, graph over time from one experiment. N = 3–5 mice per genotype. [#] Indicates *P*<0.05 on day 10 and 11 between WT/WT and α7HMZ/WT; * indicates *P*<0.01 on day 10 and 11 between α7HMZ/WT and α7HMZ/Rb KO. Right, meta-analysis of percent weight loss at 3 days of recovery after 6 days of treatment with 2.5% DSS from nine independent experiments. N = 12–47 mice per genotype. The WT/α7HMZ data include both the α7HMZ C57BL/6N parent as well as the α7HMZ C57BL/6N+J generated from the *Retnlb-/-* cross. See *Figure 6—figure supplement 1* for a comparison of the two α7HMZ lines. (Data from one experiment in which all mice, including the WT/WT controls, had lower than normal body weight loss were excluded from the analysis.) (**D**) Colon length from mice treated with 2.5% DSS for 6 days followed by

*Figure 6 continued on next page*

Figure 6 continued

different recovery periods. N = 3–14 mice per genotype per treatment. *P<0.05 versus WT/WT at different time points. #P<0.01 or ##P<0.002 versus RbKO/α7HMZ at different time points. Data are pooled from 12 independent experiments. (E) Kaplan-Meier survival curve of WT/α7HMZ (n = 4) and KO/α7HMZ (n = 9) mice after 6 days 2.5% DSS in one experiment. Meta-analysis of several independent experiments also showed that out of a total of 24 KO/α7HMZ mice allowed to go past 3 days of recovery, only one mouse died (3.6% mortality). In contrast, 13 out of 29 WT/α7HMZ mice (44.8%) either died or had to be sacrificed due to severe distress. Data for WT/α7HMZ mice in both the C57BL/6N and C57BL/6N+J lines were combined: no difference in mortality was noted between the lines. (F) WT, α7HMZ and α1HMZ mice (all in C57BL/6N background, n = 3–63-6 per genotype per treatment) were treated with 2.5% DSS for 6 days alone or followed by 3 days of recovery. Left, RELMβ protein quantified by ELISA in the midcolon homogenate: shown are means of technical triplicates from one experiment. Right, colon length. RELMβ ELISA: *P<0.03 versus WT; #P<0.01 versus α7HMZ. Colon length: *P<0.01 versus WT; #P<0.0002 versus α7HMZ. The following figure supplement is available for *Figure 6::*
The following figure supplement is available for figure 6:

**Figure supplement 1.** Verification of RELMβKO/α7HMZ mice.

## Direct and indirect mechanisms regulate RELMβ expression in α7HMZ mice

To address the mechanism responsible for increased RELMβ expression in α7HMZ mice we performed ChIP in CaCo2 cells, which express predominantly P2-HNF4α (*Chellappa et al., 2012*). We used the Support Vector Machine (SVM) learning tool in the HNF4 Binding Site Scanner to predict three potential HNF4α binding sites within 1.5 kb of the transcription start site (+1) of human *RETNLB* (*Figure 7A*, left and *Figure 7—figure supplement 1A-B*), two of which are in the vicinity of NFκB and CDX2 binding sites (*Wang, 2005*; *He et al., 2003*). We found that endogenous HNF4α binds the two regions that encompass the SVM sites (Region 1 and 2) (*Figure 7A*, right). The mouse *Retnlb* promoter also contains predicted HNF4α binding sites close to +1, one of which is highly conserved in human (*Figure 7—figure supplement 1B-C*, indicating that RELMβ expression may be directly regulated by P2-HNF4α in both mouse and human. Luciferase assays in LS174T goblet-like cells with RELMβ reporter constructs containing HNF4α binding sites in ChIP region 2 (*Figure 7—figure supplement 1D*) confirmed that P2-HNF4α activates the RELMβ promoter significantly more than P1-HNF4α (*Figure 7B*). siRNA knockdown of endogenous P1- and P2-HNF4α in LS174T cells also showed a greater effect of loss of endogenous P2-HNF4α on RELMβ expression than P1-HNF4α (*Figure 7—figure supplement 1E*). In contrast, on a known HNF4α-responsive promoter, ApoB, P1-HNF4α activates better than P2-HNF4α (*Figure 7—figure supplement 1F*).

Both P1 and P2-HNF4α are decreased after six days of DSS treatment in WT mice when RELMβ expression is increased (*Figures 2A* and *6F*), suggesting that additional mechanisms are at play in the upregulation of the *Retnlb* gene. Therefore, since RELMβ expression is known to be activated by decreased epithelial barrier function (*McVay et al., 2006*), we conducted in vivo epithelial permeability assays using Fluorescein isothiocyanate–dextran 4 kDa (FITC-dextran) and found that α7HMZ mice have moderately decreased barrier function as shown by increased FITC-dextran in the serum of both untreated and DSS-treated mice (*Figure 7C*). Furthermore, we found decreased FITC-dextran in α1HMZ mice compared to α7HMZ at 3 days of recovery (*Figure 7C*), suggesting improved barrier function, consistent with the lower levels of RELMβ and longer colon length in α1HMZ mice (*Figure 6F*). Barrier function and colon length are both indicators of colon health.

Analysis of the expression profiling data in the untreated mice revealed dysregulation of several genes related to barrier function (*Figure 7D* and *Figure 5—source data 1G*). For example, *Il4i1* and *Il13ra2,* known signaling pathways critical for RELMβ expression (*Artis et al., 2004*), are both increased in α7HMZ mice. There was also a concerted decrease in the expression of genes involved in cell adhesion and paracellular permeability in α7HMZ (*Cdh1, Cldn15, Cldn16* and *Sh3bp1*), which would contribute to decreased barrier function and hence increased DSS sensitivity and RELMβ expression in α7HMZ.

## Discussion

The majority of mammalian genes have alternative promoters, which often result in different transcript variants, but relatively little is known about the physiological relevance of those transcript

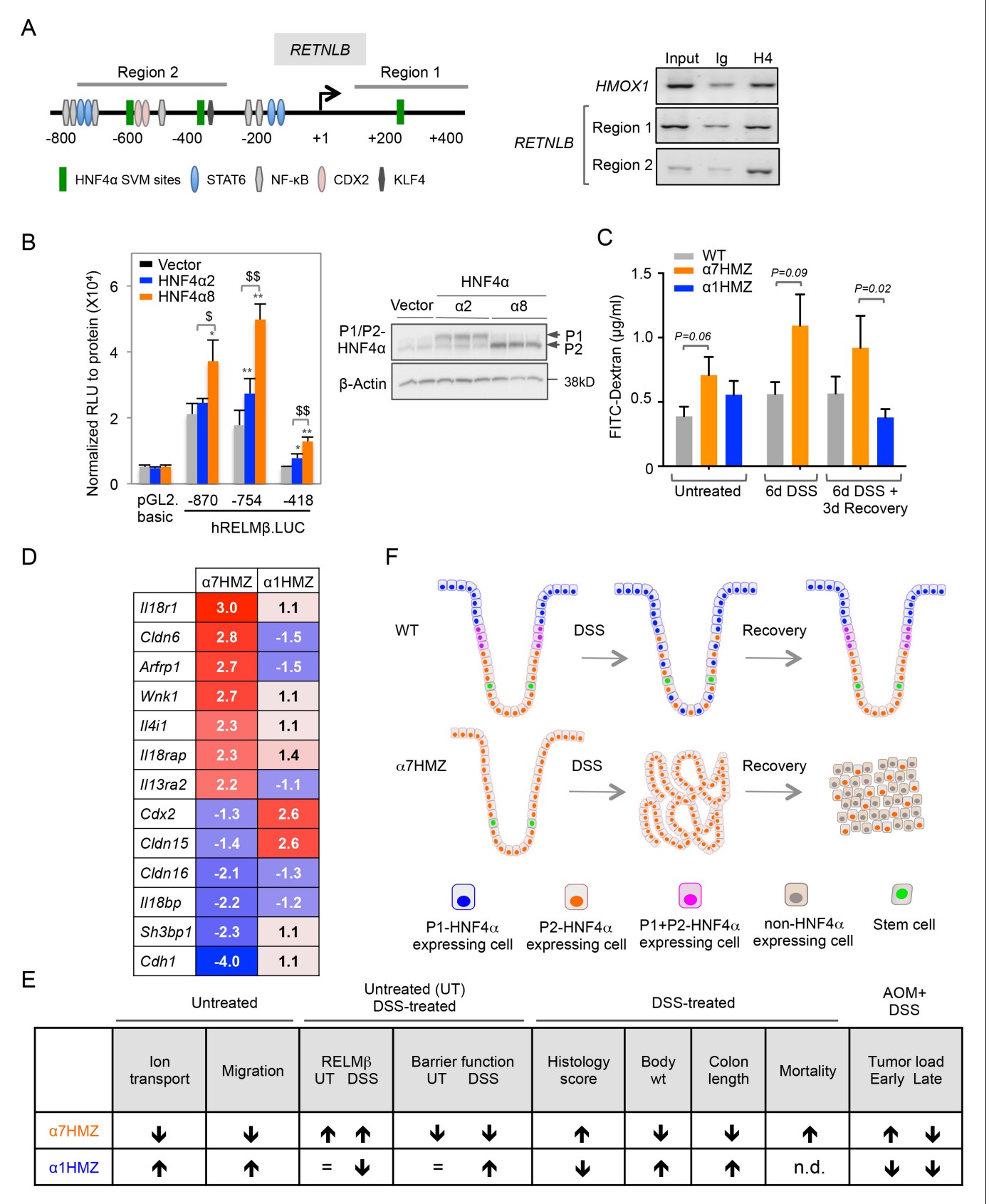

**Figure 7.** Direct and indirect mechanisms of regulation of RELMβ expression by HNF4α isoforms: impact on DSS sensitivity and recovery. (A) P2-HNF4α binds the promoter of the *RETNLB* gene in colonic epithelial cells. Left, schematic of the human *RETNLB* promoter showing predicted SVM binding sites for HNF4α, as well as sites for NFκB, KLF4, STAT6 and CDX2 (*He et al., 2003*). Right, ChIP results for endogenous HNF4α in Caco2 cells at RELMβ Region 1 and Region 2, as well as an *HMOX1* control. In, input (1/10 dilution); Ig, IgG; H4, anti-HNF4α. (B) Left, uciferase activity of pGL2.basic and RELMβ reporter constructs in LS174T cells cotransfected with vector (pcDNA3.1), human HNF4α2 or HNF4α8. Shown is the RLU normalized to protein concentration. Data are represented as mean of triplicates + *SD* of one independent experiment. \*$P<0.05$, \*\*$P<0.005$ between vector control and HNF4α2 or HNF4α8. $^{$}P<0.05$, $^{$$}P<0.005$ HNF4α2 versus HNF4α8. Right, IB of extracts from LS174T cell line cotransfected with -870.hRELMβ reporter and HNF4α isoforms. (C) Gut permeability measured by appearance of FITC dextran (4 kDa) in serum of WT, α7HMZ and α1HMZ mice either untreated, treated with 2.5% DSS for 6 days alone or followed by 3 days of recovery. (n = 7–10 for all groups except α7HMZ 6d DSS + 3d recovery where n = 4). *P*-values were determined by Student's T-test. (D) List of genes related to barrier function altered in the distal colon of α7HMZ and α1HMZ mice compared to their WT controls. Shown is nonlog fold change from the microarray experiment in *Figure 5*. (E) Summary of various phenotypes of HNFα isoform-specific mice (α7HMZ and α1HMZ) relative to WT mice in untreated, DSS and AOM+DSS treated animals as indicated. n. d., not done. =, no change. (F) Distribution of HNF4α isoforms in colonic crypts and their effect on crypt structure in DSS-induced colitis. See text for details. The following figure supplement is available for *Figure 7*:

The following figure supplement is available for figure 7:

**Figure supplement 1.** Predicted HNF4α binding sites in the human and mouse RELMβ gene and RELMβ reporter assays.

variants (*Djebali et al., 2012*; *Xin et al., 2008*). Here, we examined the HNF4α gene, which has two highly conserved promoters (P1 and P2) that give rise to proteins with different N-termini (P1-HNF4α and P2-HNF4α). Even though both promoters are expressed in adult intestines (*Tanaka et al., 2006*; *Nakhei et al., 1998*; *Briançon and Weiss, 2006*) and HNF4α has been implicated in human colon cancer (*Chellappa et al., 2012*; *Tanaka et al., 2006*; *Oshima et al., 2007*; *Cancer Genome Atlas Network, 2012*; *Zhang et al., 2014*) and colitis (*Ahn et al., 2008*; *Barrett et al., 2009*; *Fang et al., 2011*), the distribution and function of the different HNF4α isoforms in the colon have not been investigated until now (results summarized in *Figure 7E*). We show that in untreated WT adult mice P1-HNF4α protein is expressed in the differentiated compartment and the surface of the colonic epithelium whereas P2-HNF4α expression is restricted to the proliferative and differentiated compartments (*Figure 7F*). Transgenic mice expressing just a single isoform of HNF4α developed morphologically normal intestines, but hundreds of genes in the colon were differentially expressed compared to WT mice, many of which are consistent with altered barrier function and localization of the isoforms: mice expressing only P1-HNF4α (α1HMZ) had an upregulation of genes involved in differentiation, while mice expressing only P2-HNF4α (α7HMZ) had higher levels of DNA repair and cell cycle genes. Furthermore, we found that HNF4α isoforms interact in vivo with unique sets of proteins, especially those involved in signal transduction, which could contribute to differential gene expression patterns and thereby result in different susceptibilities to colitis and colitis-associated colon cancer in the isoform-specific mice. Finally, we show that *Retnlb*, which encodes the cytokine RELMβ, a key player in DSS-induced colitis, is a downstream target of P2-HNF4α.

## Role of HNF4α isoforms in colon cancer

In a mouse model of CAC, α1HMZ mice exhibited decreased tumor load, suggesting that expression of HNF4α1 from the P2 promoter in the proliferative compartment may protect against tumorigenesis, consistent with studies showing a loss of P1-HNF4α in human colon cancer (*Chellappa et al., 2012*; *Tanaka et al., 2006*; *Oshima et al., 2007*), a tumor suppressive role for P1-HNF4α in mouse liver (*Hatziapostolou et al., 2011*; *Walesky et al., 2013b*) and our recent colon cancer xenograft studies showing that ectopic expression of P1-HNF4α reduces tumor growth (*Vuong et al., 2015*). In contrast, α7HMZ mice showed an initial increase in tumorigenesis, which could reflect the absence of P1-HNF4α. Since there was no increase in the number of Ki67- or BrdU-positive cells in α7HMZ colons, no visible tumors after a chronic colitis regimen (three cycles of DSS treatment) (data not shown) nor acceleration of tumor growth in the presence of ectopic expression of P2-HNF4α in the xenograft model (*Vuong et al., 2015*), there is no indication that P2-HNF4α actively promotes proliferation. Rather, P2-HNF4α appears to be merely permissive of cell proliferation, consistent with its expression in the proliferative compartment and its retention in human colon cancer (*Chellappa et al., 2012*; *Tanaka et al., 2006*).. Interestingly, HNF4α has been shown to act as an

oncogene in gastric cancer and only P2-HNF4α is expressed in the stomach (*Chang et al., 2014*; *Dean et al., 2010*).

## Role of HNF4α isoforms in DSS-induced colitis

An additional and/or alternative explanation for the differences in CAC-induced tumors in the isoform-specific mice could be their remarkable differences in DSS sensitivity, which in turn could be due to opposing chloride secretory responses and epithelial migration (*Figure 7E*). Since the chloride secretory pathway is required for maintaining proper luminal hydration, which helps protect the epithelium from physical damage (*Barrett and Keely, 2000*), this suggests that the isoform-specific mice have different barrier functions, which we confirmed with FITC-dextran assays. Decreased expression of cell adhesion genes in α7HMZ colons – such as E-cadherin (*Cdh1*), an established HNF4α target (*Battle et al., 2006*; *Elbediwy et al., 2012*) critical for both migration of cells along crypt-villi axis and epithelial barrier function (*Grill et al., 2015*; *Schneider et al., 2010*) – would contribute to decreased barrier function. In contrast, downstream effectors of IL-18 signaling (*Il18r1* and *Il18rap*), which are implicated in intestinal epithelial barrier function (*Nowarski et al., 2015*), were increased in α7HMZ mice, while *Il18bp*, a decoy receptor for IL-18 which attenuates signaling, is decreased. All told, there are several key cell adhesion and cytokine signaling genes that are dysregulated in α7HMZ colons that could contribute to decreased barrier function and subsequently a pro-inflammatory state (*Hogan et al., 2006*; *McVay et al., 2006*), which could in turn contribute to the enhanced colitis and tumorigenesis observed in α7HMZ mice. One such cytokine is RELMβ, which we show is a direct target of HNF4α and preferentially activated by P2-HNF4α.

DSS also causes epithelial injury and a need for rapid proliferation, expansion, migration and differentiation of intestinal epithelial cells to promote wound healing and regeneration (*Sturm, 2008*). Hence, the inability of α7HMZ mice to effectively recover from DSS could be attributed to defective migration, chloride secretion (ion transport) and/or differentiation (*Figure 7F*). BrdU-labeled cells exhibited greater migration in α1HMZ and lower migration in α7HMZ mice: α7HMZ mice also had reduced expression of genes involved in cell motility. *Cldn15* is downregulated in α7HMZ and upregulated in α1HMZ colons and a known target of HNF4α (*Darsigny et al., 2009*). *Cldn15* dysregulation could explain the decrease in secretory capacity in α7HMZ mice and hence their inability to recover after DSS as a basal level of secretion is important for proper gut formation (*Anderson and Van Itallie, 2009*; *Tamura et al., 2008*). *Cdx2*, another established target of HNF4α (*Saandi et al., 2013*) and a major player in intestinal differentiation (*Suh and Traber, 1996*; *Lorentz et al., 1997*), has a similar expression profile. Finally, the involvement of these processes in recovery from DSS could explain why the crypt structure in the α7HMZ mice is not completely ameliorated by the RELMβ knockout even though body weight and colon length loss and lethality are: it has been noted previously that RELMβ expression per se does not alter colonic epithelial proliferation *McVay et al., 2006* and its role in affecting the barrier function is still debated (*Hogan et al., 2006*; *McVay et al., 2006*).

In summary, the results presented here indicate that while P1- and P2-HNF4α isoforms can substitute for each other during normal development and homeostasis, under conditions of stress they play notably different roles. Those roles seem to be driven by unique interacting partners leading to differential expression of target genes. The results also show that any factor that disrupts the balance between the HNF4α isoforms in the colon could have serious functional consequences. Those factors include Src tyrosine kinase (*Chellappa et al., 2012*), as well as any one of a number of other signaling molecules that interact preferentially with the isoforms. Future studies will be required to elucidate all the underlying mechanisms but it is anticipated that several will be important in diagnosing and treating gastrointestinal diseases involving HNF4α.

# Materials and methods

## Animal use and care

Care and treatment of animals were in strict accordance with guidelines from the University of California Riverside Institutional Animal Care and Use Committee (Protocol number A200140014). Mice were maintained in isolator cages under 12 hr light/dark cycles at ∼21°C on bedding (Andersons bed OCOB Lab 1/8 1.25CF) from Newco (Rancho Cucamonga, CA) and either fed a standard lab

chow (LabDiet, #5001, St. Louis, MO) and maintained in an open access vivarium or fed an irradiated chow (LabDiet, #5053) and housed in a specific pathogen-free (SPF) vivarium (α7HMZ and *Retnlb*⁻/⁻ matings). All experiments were performed in an open access vivarium except those in Figure 4CEF where the mice were born in an open-access vivarium and then moved to an SPF facility before treatment (due to a required institutional change). Subsequently, mice born in the SPF facility were brought to an open access facility at least two weeks prior to DSS treatment.

Transgenic mice on a mixed 129/Sv plus C57BL/6 background carrying exon 1A or exon 1D in both the P1 and P2 promoter (α1HMZ or α7HMZ, respectively) have been described previously (*Briançon and Weiss, 2006*). Both lines were maintained as heterozygotes (HTZ); wildtype (WT) and homozygous (HMZ) were mated for a single generation to generate mice for experiments. Appropriate, age-matched WT controls for both the α1HMZ and α7HMZ lines were used (designated WTα1 and WTα7, respectively, *Figures 1–5*). The α7HMZ and α1HMZ mice were backcrossed to C57BL/6N for 10+ generations and used with C57BL/6N WT controls (*Figure 6*). The backcrossed α7HMZ mice were crossed with RELMβ knockout (*Retnlb*⁻/⁻) mice which were generated as previously described (*Hogan et al., 2006*) using VelociGene technology. The *Retnlb*⁻/⁻ mice were backcrossed 6+ generations in C57BL/6J to generate RbKO/α7HMZ mice in a C57BL/6N+J background. WT/α7HMZ mice from the RELMβ cross showed essentially identical susceptibility to DSS as the α7HMZ parent in the C57BL/6N background (as well as the original exon swap mice in the mixed background) (*Figure 6—figure supplement 1C* and data not shown). All experiments with RbKO/α7HMZ mice were compared to RbWT/α7HMZ from the RELMβ cross except for the meta analysis in *Figure 6C* which included data from the parental α7HMZ line in C57BL/6N. Mice of the same genotype were housed three to five per cage, randomly assigned to treatment groups at the beginning of the experiment and subjected to a single experimental regime in their cages. Mice were euthanized by $CO_2$ asphyxiation and tissues harvested in the mid morning to mid afternoon.

## DSS colitis

Male mice (10 to 16 weeks old) were treated with 2.5% DSS (MW 36,000–50,000 Da, MP Biomedicals, #160110, Santa Ana, CA) in water given ad libitum for four to six days and sacrificed immediately or allowed to recover up to 18 days with tap water. WT mice were treated in parallel in each experiment as controls for the DSS: the same lot number of DSS was used for a given group of experiments whenever possible to avoid lot-to-lot variation. Mice in severe distress (weighing 13 grams or less, or excessively hunched and lethargic) were euthanized prior to the termination of the experiment except in experiments measuring mortality. To avoid confounding effects due to unrelated illnesses, when an animal became unexpectedly ill, all mice in the cage were excluded from the analysis.

## Colitis-associated and sporadic colon cancer

CAC was established as described (*Neufert et al., 2007*). Briefly, we intraperitoneally (i.p.) injected male mice (6 to 10 weeks old) with 10 mg/kg AOM (National Cancer Institute, Bethesda, MD) on Day 1 in the morning. On Day 2 mice were given 2.5% DSS in water for four to seven days, followed by 16 days of untreated water; the cycle was repeated one or two additional times. Mice were sacrificed at day 46 to 95; tumor number counted by visual inspection and tumor size measured with digital calipers were determined in a blind fashion. Sporadic colon cancer was induced in male mice (6 to 8 weeks old) by i.p. injection of 10 mg/kg AOM once a week for six consecutive weeks. Mice were sacrificed 28 weeks after the first injection.

## H&E staining and immunofluorescence

Distal colons were fixed in 10% phosphate buffered formalin and stained with hematoxylin and eosin (H&E) or for immunofluorescence (IF) as described previously (*Tanaka et al., 2006*; *Lytle et al., 2005*). For antigen retrieval, tissue sections were soaked in 1% SDS in phosphate buffered saline (PBS) and microwaved for 2 min for all antibodies except for P2-HNF4α which was autoclaved at 121°C for 20 min in 10 mM citrate buffer. Images were captured with a Zeiss 510 confocal microscope. Mouse monoclonal antibodies to P1/P2-driven HNF4α (#PP-H1415-00), P1-driven HNF4α (#PP-K9218-00) and P2-driven HNF4α (#PP-H6939-00) were from R&D Systems (Minneapolis, MN). Antibodies to Ki67 were from Abcam (#ab1667, Cambridge MA). Rabbit NKCC1 antibody (TEFS2)

has been described previously (*McDaniel and Lytle, 1999*). Alexa fluor anti-mouse and anti-rabbit secondary antibodies and TO-PRO-3 nuclear stain (red) were from Life Technologies (Carlsbad, CA). For RELMβ IF staining, antigen retrieval was performed by immersion of slides in 95–100°C pre-heated sodium citrate buffer (10 mM). Following cooling to room temperature, slides were rinsed twice with PBS/0.1% Tween 20 for 5 min and then blocked with 5% normal donkey serum (Jackson Immuno Research Labs, Westgrove, PA) in StartingBlock (Thermo Scientific, Carlsbad, CA). Sections were stained with rabbit anti-RELMβ antibody (#500-P215, Peprotech, Rocky Hill, NJ), followed by fluorochrome-conjugated anti-rabbit antibody (Abcam), and counterstaining with DAPI (Cell Signaling Technology, Danvers, MA).

## Immunoblot
Whole cell extracts (WCE) were prepared from either snap-frozen or fresh tissue using ice-cold Triton lysis buffer by motorized (Wheaton, Millville, NJ) or manual homogenization. Triton lysis buffer was 20 mM Tris pH 7.5, 150 mM NaCl, 10% glycerol, 1% NP40, 1% Triton-X-100, 1 mM EDTA, 2 mM EGTA plus inhibitors (1 µg/ml of aprotonin, leupeptin and pepstatin, 1 mM of sodium orthovanadate, sodium fluoride and (PMSF), phosphatase inhibitor cocktail I & II (1:100), protease inhibitor cocktail (1:10 – 1:100), Sigma-Aldrich, St. Louis, MO). Protein extracts ($\sim$ 20–100 µg) were analyzed by 10% SDS-PAGE followed by transfer to Immobilon (EMD Millipore, Billerica, MA) before staining with antibodies or Coomassie for protein loading.

## Histological scoring
Blinded histology scoring of H&E stained sections was performed according to three criteria. Crypt damage: 0 = intact crypts, 1–2 = loss of basal area, 3–4 = entire crypt loss with erosion, 5 = confluent erosion. Leukocyte inflammation: 0 = no inflammatory infiltrate, 1 = leukocyte infiltration in the lamina propria, 2 = leukocyte infiltration extending into the submucosa, 3 = transmural and confluent extension on inflammatory cells. Ulceration: 0 = no ulcers, 1–2 = presence of ulcers, 3 = confluent and extensive ulceration.

## BrdU labeling
Young adult male mice were injected i.p. with 75 mg/kg BrdU (BD Biosciences, #550891, San Jose, CA) and sacrificed after 2 to 3 hr, 25 hr or 48–50 hr. Distal colons fixed in formalin were sectioned and immunostained with BrdU antibody as per manufacturer instruction (BD Biosciences, #550803). Images were captured at 40X (Zeiss Axioplan, Jen Germany) and crypt dimensions were measured using SPOT Imaging software (Sterling Heights, MI).

## Isolation of mouse colonic crypts
Distal colons from 19-week-old male WT mice from the mixed background (WTα7) were rinsed in PBS, placed in a 4% bleach solution for 20 min, washed three times in PBS and then incubated with 3 mM EDTA, 0.5 mM (DTT) in PBS for 90 min at 4°C followed by a PBS wash. Colonic crypts were isolated by vigorous shaking; they were fixed and immunostained as described (*Chellappa et al., 2012*).

## Ussing chamber assay
The short circuit current ($I_{sc}$) and electrical resistance across the mucosal layer of mouse distal colon was measured using an Ussing chamber as described (*Bajwa et al., 2009*). Electrogenic $Cl^-$ secretion was recorded as the $I_{sc}$ evoked by sequential addition of 100 µM carbachol and 20 µM forskolin (Sigma-Aldrich) to the serosal bath.

## Expression profiling
Mouse Exon 1.0 ST Arrays (Affymetrix, Santa Clara, CA) were hybridized at the University of California Riverside Genomics Core using polyA+ RNA extracted from the distal colon of young adult male mice (12 to 16 weeks old) fed a standard lab chow ad libitum in an open access vivarium. RNA was pooled from two to three mice per genotype and applied to one array; a second array was processed in a similar fashion for a total of four to six mice assayed per genotype. Results from the two arrays were averaged. Isoform-specific mice were compared to their appropriate, age-matched WT

controls. Data were analyzed using Robust Multi-array Average (RMA) background adjustment and quantile normalization on probe-level data sets with Bioconductor packages, Exonmap, and Affy software. To determine the differentially expressed transcripts only the probes with $P<0.05$ (Student's $t$-test) and more than two-fold change were considered. Gene Ontology (GO) overrepresentation analysis was conducted using DAVID. Microarray data have been deposited in the Gene Expression Omnibus MIAME-compliant database (Accession number GSE47731) (http://www.ncbi.nlm.nih.gov/geo/query/acc.cgi?acc=GSE47731).

## RELMβ ELISA

Colon tissue (~1 cm) was weighed and homogenized in 0.5 mL PBS, followed by ELISA with capture and detection biotinylated antibodies for anti-RELMβ (Cat #500-P215Bt, Peprotech) according to the manufacturer's instructions. Samples were compared to a serial-fold dilution of recombinant mouse RELMβ protein (#450-26B, Peprotech) and calculated as ng per gram tissue. All ELISAs were performed in technical triplicates.

## Chromatin immunoprecipitation (ChIP) followed by PCR

Human colonic epithelial cells Caco2 cells (ATCC HTB-37) were grown in DMEM (Dulbecco's modified Eagle's medium with 4.5 g/liter glucose, L-glutamine, and pyruvate) supplemented with 20% fetal bovine serum (FBS) (BenchMark; cat#100–106) and 100 U/mL penicillin-streptomycin (1% P/S) at 37°C and 5% $CO_2$. At ~95% confluency the cells were crosslinked with formaldehyde and subjected to ChIP analysis using the affinity purified anti-HNF4α antisera (α445), which recognizes the very C-terminus of both P1- and P2-HNF4α, as described previously (*Vuong et al., 2015*). The following primers in the *RETNLB* promoter were used in the PCR for 40 cycles: Region 1 forward 5'-CTCCTCCACCTCTTCCTCCT-3' and reverse 5'-CATCCTAATCCCCCTTCTCC-3' (301 bp); Region 2 forward 5'-CCTTTGCTCTGGATCTCTGC-3' and reverse 5'- ATGAGCCCCCAAAAGAACTC-3' (405 bp). Primers in the *HMOX1* promoter were used as a positive control, forward 5'-CCTCTCCACCCCACACTGGC-3' and reserve 5'-GCGCTGAGGACGCTCGAGAG-3' (179 bp). Primers were designed using the UCSC genome browser (https://genome.ucsc.edu/) and Primer3 (v.0.4.0) (http://bioinfo.ut.ee/primer3-0.4.0/). Predicted HNF4α binding sites (Support Vector Machine, SVM, algorithm) were identified using the HNF4α Binding Site Scanner (http://nrmotif.ucr.edu/) (*Bolotin et al., 2010*).

## Fluorescein isothiocyanate–dextran (FITC-dextran) assay

Intestinal epithelial permeability was assessed by measuring the appearance of FITC-dextran (FD-4, Sigma) in mouse serum as described previously (*Brandl et al., 2009*). Untreated or treated (2.5% DSS for six days or 2.5% DSS for six days, followed by days of recovery with tap water) WT, α7HMZ or α1HMZ mice were fasted overnight and then gavaged with FITC-dextran (60 mg/100 g body weight) 4 hr before harvesting. Blood was collected either from the inferior vena cava or by cardiac puncture and allowed to sit on ice for 30 min. Serum was collected after centrifuging the blood for 15 min at 2000 xg at 4°C. FITC-dextran measurements were performed in duplicate or triplicate by fluorometry at 490 nm. Data were analyzed using Prism 6 software (GraphPad Prism version 6 for Mac, GraphPad Software, La Jolla, CA); outliers were identified by the ROUT method and removed.

## Plasmids and siRNA

Human HNF4α2 (NM_000457) and HNF4α8 (NM_175914.3) constructs in pcDNA3.1 (+) vector were gifts from Dr. Christophe Rachez (Pasteur Institute, Paris, France) as described previously (*Chellappa et al., 2012*). pGL2.basic, human RELMβ reporter constructs and LS174T cells were gifts from Dr. Gary Wu (*Wang et al., 2005*).

ON-TARGET siRNA targeting P1- and P2-HNF4α were custom synthesized from Dharmacon. si P1-HNF4α: Sense, 5'-U U G A G A A U G U G C A G G U G U U U U -3'; Antisense 3'-U U A A C U U C U U A C A C G U C C A C A A -(5'-P) 5'. siP2-HNF4α: Sense, 5'-G U G G A G A G U U C U U A C G A C A U U-3'; Antisense, 3'-U U C A C C U C U C A A G A A U G C U G U-(5'-P) 5'. ON-TARGET-plus Non-targeting siRNA #1 (D-001810-01-20) was used as siControl.

## Luciferase assay

LS174T cell lines were grown in DMEM supplemented with 10% FBS and penicillin-streptomycin (1% P/S) and maintained at 37°C and 5% $CO_2$. For siRNA experiments, $8 \times 10^6$ LS174T cells were plated in 60-mm plates and transfected 24 hr after plating with 100 nM siRNA using RNAi Max (Invitrogen) according to the manufacturer's protocol. Forty-eight hours after transfection, cells were split into a 24-well plate ($8 \times 10^6$ cells per well), and 24 hr later transfected with Lipofectamine 3000 according to the manufacturer's protocol (Invitrogen) with CMV.βgal (50 ng) and pGL2.basic or human RELMβ reporter constructs (1 μg). For HNF4α transfections, $8 \times 10^6$ LS174T cells were plated in 24-well plates and 24 hr later transfected with human HNF4α2 or HNF4α8 (100 ng), CMV.βgal (50 ng) and pGL2. basic or human RELMβ reporter constructs (1 μg). Cells were harvested 24 hr after transfection using passive lysis buffer (Promega). Luciferase and β-gal activity were measured as described previously (*Chellappa et al., 2012*).

## RIME (Rapid Immunoprecipitation Mass spectrometry of Endogenous proteins) for mouse colon

RIME was carried out as previously described (*Mohammed et al., 2013*), with the following modifications. Whole colon from α1HMZ and α7HMZ untreated male mice (n = 3 per genotype, ~16 weeks of age, backcrossed into C57/BL6N and maintained in an SPF vivarium) were fixed in 1.1% methanol-free formaldehyde (5 mL) [in 1 x phosphate buffered saline (PBS plus 1 mM PMSF and DTT, 2 μg/mL leupeptin and aprotinin, and 1:200 protein phosphatase inhibitors I (2&3) (Sigma)] for 10 min at RT; crosslinking was then stopped with 0.125 M glycine for 5 min at RT. Fixed colon was further processed into single cells at 4°C. Colon was lightly minced with a razor blade and disaggregated in PBS plus inhibitors (as above) by passing through a motorized homogenizer. The cells were then drained through a cell strainer and dounced using a hand-held homogenizer. Cells were swelled in 1.0 mL Hypotonic Buffer (10 mM HEPES-KOH pH 7.9, 10 mM KCl, 1.5 mM $MgCl_2$) plus inhibitors for 10 min at 4°C and centrifuged to collect the nuclei. The pellet was washed with Nuclei Buffer (1% SDS, 50 mM Tris-Cl pH 8.0, 10 mM EDTA) plus inhibitors and D3 Buffer (0.1% SDS, 10 mM Tris-Cl, 1 mM EDTA). The pellet was resuspended in fresh 0.5 mL D3 Buffer plus inhibitors, transferred to a 1.0 mL AFA millitube with a plastic stirring rod and more D3 Buffer was added to fill the tube (total volume ~780 μL). Samples were sonicated for 9 min (4 min per 200 cycles/bursts, 5.0 duty force, and 140 peak power; one min delay) in a Covaris S220, Bioruptor and pre-cleared for 30 min in 10 μL of magnetic beads (Pierce Thermo Scientific, cat#88802). Prior to use, magnetic beads were washed 3 x 1.0 mL in cold PBS. Samples were split in half and diluted 1:1 with Immunoprecipitation (IP) Dilution Buffer (0.01% SDS, 20 mM Tris-Cl pH8.0, 1.1% Triton X-100, 167 mM NaCl, 1.2 mM EDTA) and placed in non-stick tubes. Each half sample was incubated with 40 μL of magnetic beads that were pre-incubated with 21 μg of anti-HNF4α (α-445) (*Sladek et al., 1990*) or rabbit IgG in 0.05% Tween PBS at 4°C overnight. The following day, IPs were washed 3 x 1.0 mL with ice cold Radioimmunoprecipitation assay (RIPA) Buffer (15 mM Tris-Cl, 150 mM NaCl, 1% NP-40, 0.7% deoxycholate), washed 1 x 400 μL with DNaseI Buffer (40 mM Tris-Cl, 1 mM $CaCl_2$, 10 mM NaCl, 6 mM $MgCl_2$) and incubated in 100 μL DNaseI Buffer and 8 μL of DNaseI enzyme (4 μg/μL) for 20 min at 30°C. Additional washes were done (3 x 0.5 mL with RT RIPA Buffer and 2 x 1.0 mL with cold RIPA Buffer). IP'd material was washed twice with cold 50 mM $NH_4CO_3$. At the last wash (0.5 mL), samples were transferred to new tubes. Wash buffer was removed and IP beads subjected to mass spectrometry as follows:

Sample beads were washed with trypsin digestion buffer, digested with trypsin overnight and subjected to 2D-nanoLC/MS/MS analysis at the UCR Institute of Integrated Genome Biology Proteomics Core as described previously (*Drakakaki et al., 2012*). Briefly, a MudPIT approach employing a two-dimension nanoAcquity UPLC (Waters, Milford, MA) and an Orbitrap Fusion method (Thermo Scientific, San Jose, CA) was used to analyze all sample. Data were acquired using Orbitrap fusion method (*Hebert et al., 2014*) with acquisition time set from 1–70 min. For MS2 scanning only precursor ions with intensity of 50,000 or higher were selected and scanned from most intense to least intense precursor ions within 4 s. A 5-s exclusion window was applied to all abundant ions to avoid repetitive MS2 scanning on the same precursor ions using 10 ppm error tolerance. All raw MS data were processed and analyzed using Proteome Discoverer version 2.1 (Thermo Scientific, San Jose, CA). Only proteins with 1% FDR cut-off (q≤0.01) were considered for subsequent analysis.

Proteins had to be present in at least two out of the three replicates with the HNF4$\alpha$ antibody and not in any IgG control in both $\alpha$1HMZ and $\alpha$7HMZ samples to be considered for the 'both' category. To be considered in the '$\alpha$7HMZ only' category, the protein had to appear either in two of three $\alpha$7HMZ samples but in none of the three $\alpha$1HMZ samples, or in three of the $\alpha$7HMZ samples and only one of the $\alpha$1HMZ samples. A similar strategy was used for the '$\alpha$1HMZ only' proteins. Proteins were converted to gene symbols and cross-referenced with human and mouse genes associated with colon cancer, IBD, Crohn's disease and ulcerative colitis found in the literature (*Franke et al., 2010*; *Jostins et al., 2012*) and a Pubmed-Gene search conducted in April 2016, followed by manual curation. Gene Ontology using Panther (www.pantherdb.org) as well as manual curation resulted in the TF, RNA binding and protein kinase and phosphatase categories in *Figure 5C* bottom. The Human Protein Atlas (http://www.proteinatlas.org/) was used to confirm expression in the colon and nucleus.

## Statistical analyses

Sample sizes for DSS and AOM+DSS regimes were determined on the basis of mouse-to-mouse variation in body weight loss and tumor number/load (respectively) observed in pilot experiments. Each mouse was considered to be a biological replicate; technical replicates refer to multiple analyses of the same tissue from a given animal. All results are expressed as the mean $\pm$ s.e.m, of sample size $n$. Significance was tested by analysis of variance or Student's $t$-test. Probabilities less than 5% ($P<0.05$) were considered to be significant. For RIME, a cut off of $q<0.01$ (1% FDR) was used.

## Acknowledgements

We thank J Schnabl, B Wang, A Mamaril and H Evans for technical assistance, S Pan for mass spec analysis, JR Deans for bioinformatics analysis of RIME, G Wu for RELM$\beta$ reporter constructs and LS174T cell line and MC Weiss for the exon swap mice.

## Additional information

### Funding

| Funder | Grant reference number | Author |
|---|---|---|
| National Institute of Diabetes and Digestive and Kidney Diseases | R01DK053892 | Frances M Sladek |
| National Institute of Environmental Health Sciences | 5T32ES018827 | Poonamjot Deol |
| National Institute of Allergy and Infectious Diseases | R01AI091759 | Meera G Nair |

The funders had no role in study design, data collection and interpretation, or the decision to submit the work for publication.

### Author contributions

KC, FMS, Conception and design, Acquisition of data, Analysis and interpretation of data, Drafting or revising the article; PD, JRE, LMV, GC, Acquisition of data, Analysis and interpretation of data, Drafting or revising the article; NB, Drafting or revising the article, Contributed unpublished essential data or reagents; EB, MGN, Analysis and interpretation of data, Drafting or revising the article; CL, Analysis and interpretation of data, Drafting or revising the article, Contributed unpublished essential data or reagents

### Ethics

Animal experimentation: Care and treatment of animals were in strict accordance with guidelines from the University of California Riverside Institutional Animal Care and Use Committee. Institutional protocol number A200140014.

## Additional files

### Supplementary files
• Repoarting Standard. NC3Rs ARRIVE guidelines checklist.

### Major datasets
The following dataset was generated:

| Author(s) | Year | Dataset title | Dataset URL | Database, license, and accessibility information |
|-----------|------|---------------|-------------|--------------------------------------------------|
| Karthikeyani Chellappa, Eugene Bolotin, Frances M Sladek | 2013 | Differential role of HNF4alpha isoforms in colitis and colitis-associated colon cancer | http://www.ncbi.nlm.nih.gov/geo/query/acc.cgi?acc=GSE47731 | Publicly available at NCBI Gene Expression Omnibus (accession no: GSE47731) |

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
