## [Decision Letter]

Thank you for submitting your work entitled "Opposing roles of nuclear receptor HNF4α isoforms in colitis and colitis-associated colon cancer" for peer review at *eLife*. Your submission has been favorably evaluated by Kevin Struhl (Senior editor) and an additional expert reviewer.

The reviewers have discussed the reviews with one another and the Reviewing editor has drafted this decision to help you prepare a revised submission.

Both reviewers think that the paper is interesting and potentially appropriate for publication in *eLife*. However, though nicely demonstrated, the conclusion that the P1 form is tumor suppressing and the P2 form is oncogenic is insufficient for publication in *eLife*. What is missing is a mechanistic connection between the 2 isoforms and expression of RELMβ. Thus, publication in *eLife* requires making such a mechanistic connection, particularly showing that the P2 isoform, but not the P1 isoform, activates the RELMβ promoter through HNF4α sites. There are several possible options (reporter assays, ChIP, mutation of the site) that could be completed in the standard 2-month period (or slightly longer) since they are standard experiments. Ideally, it would be nice to show that P2-mediated regulation of RELMβ per se is critical in mice as mentioned by Reviewer 1, but this is likely to take too long and is not required. For your information, the full reviews are included below.

*Reviewer #1:*

This paper presents useful data, but it belongs in a more specialized journal. There is already a considerable body of evidence indicating that the HNF4α isoforms are differentially regulated and have different functions. In particular, there is considerable evidence that the P1 form acts as a tumor suppressor. Also, over-expression of HNF4α is oncogenic despite the P1 form, suggesting that P2 acts as an oncogene. The authors nicely test this by generating mice that can only express either the P1 or P2 isoforms and assessing various phenotypes including gene expression profiles and colitis susceptibilities, the authors significantly improve the understanding of the distinct roles of these isoforms. To this non-expert, the experiments generally seem fine with clear results. However, the differences between the isoforms are phenomenological with limited mechanistic understanding, and their opposing roles in colitis and colitis-associated colon cancer are in accord with previous characterization as tumor suppressing and tumor promoting. As such, I think this paper will be of interest to specialists in HNF4α and colitis/colon cancer, but it is unclear what findings are of more general significance.

It is already known that loss of RELMβ causes resistance to colitis. The new finding here is this gene is up-regulated by the P2 form, which of course also stimulates colitis. So, the knock-out result is largely expected. Moreover, the conclusion that P2-mediated up-regulation of RELMβ is important is not really tested. The correct experiment would be to knock out HNF4 sites that are required for the up-regulation of RELMβ and show that this blocks colitis. Alternatively, some other manipulation that reduces RELMβ levels to the uninduced level. To put it differently, one has to show that the regulation of RELMβ is important, not just the gene itself which is already known. I realize this is a fair amount of work.

*Reviewer #2:*

HNF4α expression in the gut has been linked to both colitis and colon cancer. Chellapa et al. describe strikingly differential effects of distinct isoforms of HNF4α in the gut. Expression of the P2 promoter isoform only results in increased sensitivity to DSS and increased tumorigenesis in an AOX/DSS model, while opposite results are observed with mice that express only the P1 isoforms. Within the context of nuclear receptor function, it is well known that there are multiple isoforms of many NRs, but they are generally not distinguished functionally. Thus, this represents a particularly clear example of delineation of distinct functions of different isoforms. This observation might be of more limited interest, but clear mechanistic information is provided by the demonstration that RELMβ is overexpressed in the DSS sensitive P2-specific mice, and that doubly mutant P2 specific/RELMβ knockout mice are resistant to the increased sensitivity to DSS.

One question raised by these results is whether the effects of the different isoforms on RELMβ expression are direct or indirect, and particularly whether the distinct isoforms have distinct direct effects on RELMβ expression. This study would be further strengthened if relatively straightforward RELMβ promoter cotransfections showed such effects, or if other strategies could document differential promoter occupancy.

[Editors’ note: what now follows is the decision letter after the authors submitted for further consideration. The paper was rejected in the second round of review, but then accepted after the authors appealed against the decision.]

Thank you for submitting your work entitled "Opposing roles of nuclear receptor HNF4α isoforms in colitis and colitis-associated colon cancer" for consideration by *eLife*. Your article has been reviewed by one of the original peer reviewers, and the evaluation has been overseen by Kevin Struhl as the Senior Editor, who also provided a full critique. Our decision has been reached after consultation between the reviewers. Based on these discussions and the individual reviews below, we regret to inform you that your work will not be considered further for publication in *eLife*.

The two re-reviews are very similar. While the new experiments improve the manuscript, they don't address the key question that was required, namely no mechanistic connection between the 2 isoforms, expression of RELMβ, and the phenotype. Perhaps the authors could perform an experiment (one suggested below) that would address this, but at this point the paper is not acceptable. If the authors do such an experiment, it would be reasonable to reconsider on appeal.

*Reviewer #1:*

The authors have provided thoughtful responses to my comments, and they have done some new experiments that are useful. Nevertheless, I don't think they fully understood my point of view, and they only minimally addressed the key issue.

1) The paper clearly represents an advance over published work, the experiments are well done, and it deserves publication. I don't dispute any of the authors' responses about how their paper is advanced over previous work. However, the relevant question is how significant the advance is over previous work, particularly at the "general interest" level needed for publication in *eLife*. From my outside perspective, there is already evidence (perhaps considerable was too strong) that P1 acts as a tumor suppressor and that P2 acts an oncogene (less explicitly demonstrated, but the only reasonable way to explain how overproduction of HNF4α is oncogenic). The authors experiments are a nice and more explicit demonstration of the opposing roles of the P1 vs P2 forms, but I do not view this is as a major advance. Yes, the experimental systems are different and the experiments better, but the basic point is the same. For this reason, I (and the other Reviewer) thought that the basic opposing phenotypes were worth publishing, but not a sufficient advance for *eLife*.

2) Because of point 1, I viewed it as important to have some mechanistic understanding of how the distinct HNF4α isoforms are linked to RELMβ. Of course, the primary finding (point 1) means that mice expressing these different isoforms have very different phenotypes and hence very different transcriptional profiles. Ultimately, this means that the different isoforms have different transcriptional effects, but profiling data per se provides no mechanistic information. Conversely, the previous knowledge that RELMβ is linked to colitis combined with the authors' finding that RELMβ levels are increased in P2 mice does not indicate that this up-regulation is functionally important. It is entirely possible that a RELMβ derivative expressed at its normal level but unable to be induced by P2 would behave similarly to the wt (inducible) allele. To put it differently, one predicts that increased DSS sensitivity would depend on RELMβ no matter how once increased DSS sensitivity is achieved. So, again, there is no specific link between P2-overexpression and the RELMβ requirement for DSS sensitivity.

3) The new ChIP experiments are useful, because they show that, in principle, the HNF4α isoforms could directly affect transcription of RELMβ. However, this does not really address the critical point, namely the differential behavior of the isoforms. Also, the focus on DNA-binding is possibly misplaced, as it is entirely possible that the distinction between the isoforms relates to transcriptional potential or interaction with other proteins (with differential transcriptional potential). I agree that artificial reporters with HNF4α sites is likely to be useless. But, what I imagined was an experiment with a reporter driven by a reasonably large segment of the RELMβ locus and co-transfection of the individual isoforms.

4) The new barrier function experiments are nice on their own, but don't address mechanism related to the distinct isoforms. Again, the P1 vs. P2 cells have many differences, but those differences per se don't address any mechanistic understanding. The effects on barrier function and/or RELMβ could be very indirect consequences of the distinct HNF4α isoforms. By definition, the phenotypic differences caused by the distinct isoforms means that, at some level any property (molecular or overall phenotype) is differentially affected. In addition, the relationship of the P2 mice vs. other versions of DSS-induced colitis means that there are mechanistic similarities between those situations, but they don't address be basis of why P1 vs. P2 are different.

For these reasons, although the manuscript is improved, I still believe that the differences between the isoforms are phenomenological with limited mechanistic understanding, and their opposing roles in colitis and colitis-associated colon cancer are in accord with previous characterization as tumor suppressing and tumor promoting. As such, I think this paper will be of interest to specialists in HNF4α and colitis/colon cancer, and certainly worth publishing in a specialized journal, but it is unclear what findings are of more general significance.

*Reviewer #2:*

This revised manuscript has been improved by the direct demonstration that RELMβ is an HNF4α target gene, and also the identification of the differential barrier function in the two isoform specific strains. However, the primary requirement for acceptance of this manuscript was clearly stated in the reviewing editor's decision: "What is missing is a mechanistic connection between the 2 isoforms and expression of RELMβ. Thus, publication in *eLife* requires making such a mechanistic connection…". As the authors acknowledge, this is a difficult, but important issue. Thus, despite the improvements, this revised manuscript has not effectively answered the primary question.

---

## [Author Response]

[Editors’ note: the author responses to the first round of peer review follow.]

Both reviewers think that the paper is interesting and potentially appropriate for publication in eLife. However, though nicely demonstrated, the conclusion that the P1 form is tumor suppressing and the P2 form is oncogenic is insufficient for publication in eLife. What is missing is a mechanistic connection between the 2 isoforms and expression of RELMβ. Thus, publication in eLife requires making such a mechanistic connection, particularly showing that the P2 isoform, but not the P1 isoform, activates the RELMβ promoter through HNF4α sites. There are several possible options (reporter assays, ChIP, mutation of the site) that could be completed in the standard 2-month period (or slightly longer) since they are standard experiments. Ideally, it would be nice to show that P2-mediated regulation of RELMβ per se is critical in mice as mentioned by Reviewer 1, but this is likely to take too long and is not required. For your information, the full reviews are included below.

To reiterate the main finding of our work, we report for the first time that HNF4α isoforms driven by alternate promoters regulate colitis and colitis-associated colon cancer in an isoform-specific manner. We illustrate that P1-HNF4α attenuates, whereas P2-HNF4α accelerates, colitis and colitis-associated colon cancer, respectively. Our findings emphasize the importance of tracking the specific changes in P1- and P2-HNF4α isoforms under multiple pathological conditions, including inflammatory bowel disease (IBD) and colon cancer.

(We would also like to re-emphasize a more general relevance of our work. Many, if not most, human genes have more than one promoter and often those alternative promoter structures are conserved between human and mouse, as is that of HNF4α. And yet there is very little if any in vivo work on the effects of those alternative promoters. Our use of the HNF4α exon swap mice provides the ideal system with which to examine the effect of the different isoforms under physiological conditions of expression.)

In addition to elucidating an in vivo distinction between the HNF4α isoforms, we also identify the major mechanismby which α7HMZ mice have increased susceptibility to DSS-induced colitis – namely that the mice have increased expression of the cytokine Relmβ – and we provide strong proof of that mechanismby making double transgenic mice (RELMβ KO/α7HMZ). In this sense, we have already provided the mechanism for the main point of our paper but we understand the reviewers’ desire to dig deeper and link the expression of Relmβ directly to the P2-HNF4α isoform (expressed in α7HMZ mice) but not the P1-HNF4α isoform.

While at first glance the simplest explanation may seem to be that HNF4α binds the RELMβ promoter in an isoform-specific manner, there are several reasons why this might not be the case. The DNA binding domains of P1- and P2- HNF4α are 100% identical (as indicated in Figure 1) and we have already shown using a high throughput DNA binding assay in which we examined 250,000 unique sequences, as well as ChIPseq in HCT116 cells, that P1 and P2- HNF4α bind DNA with a very similar specificity both in vitroand in vivo(Vuong et al. 2015 MCB, PMID: 26240283) (We have also found nearly identical ChIPseq profiles of HNF4α in the livers of the isoform-specific mice, unpublished).

Nonetheless, it is true that we have observed some differences between the isoforms in DNA binding both in vitroand in vivo, so could the Relmβ promoter be one of those rare examples?

To address this issue we took two different approaches to examine both direct and indirect mechanisms responsible for the regulation of RELMβ by the HNF4α isoforms.

1) Direct regulation of RELMβ expression by HNF4α

There are two reasons why we did not initially investigate further the possibility of HNF4α directly regulating the expression of the RELMβ gene. One, we did not detect any HNF4α ChIPseq peaks (or RNAseq signal) in the RELMβ promoter in the HCT116 cell line expressing either HNF4α2 (P1 isoform) or HNF4α8 (P2 isoform) (PMID: 26240283). Two, Verzi’s group did not detect ChIPseq peaks for HNF4α in the CaCo2 cell line (PMID: 21074721). However, these negative results could be due either to the cell type (HCT116 cells do not express endogenous HNF4α and are considered to be stem cell like while Caco2 cells are a cancer cell line, albeit of an epithelial cell origin and they express endogenousl HNF4α) and/or technical issues (especially in terms of the Caco2 cells since we did not perform those experiments). The other possibility is that, in addition to HNF4α, other factors are needed to activate the expression of the RELMβ gene, such as CDX2, NFkB and STAT pathways, which are activated in response to bacteria (PMID:15576623). Neither the Caco2 nor the HCT116 cells were exposed to exogenous insults like LPS or cytokines for the ChIPseq experiments.

Therefore, prompted by the reviewer’s comment, we ran the RELMβ promoter through the HNF4 Binding Site Scanner developed by our lab and identified several potential SVM sites (SVM = support vector machine learning, an algorithm that uses 1000’s of experimentally verified binding sites for HNF4α to predict potential new binding sites (see Bolotin et al. 2010 Hepatology PMID:20054869 and http://nrmotif.ucr.edu/). We chose two regions to focus on which we refer to RELMβ Region 1 and Region 2. Region 1 has a high scoring SVM site (score of 1.94) and Region 2 has two lower scoring SVM sites (~1.3). While typically we consider sites with scores of 1.5 or higher as good binders, we have also observed sites with lower scores (anything over 1.0) as binders as well. Region 2 also overlaps with NFkB, KLF4 and CDX2 sites, which have been shown to be involved in the activation of the RELMβ promoter in response to LPS in CaCo2 cells [PMID:15576623].

We performed HNF4α ChIP assays in the CaCo2 cell line and found that HNF4α binds the RELMβ promoter at both Region 1 and 2 (new Figure 7). We have previously shown that the P2-isoform represents the majority of HNF4**α** expressed in Caco2 cells (~90%), while the P1- isoform is expressed at a very low level (~10%). (Chellappa et al. 2012 PNAS, PMID: 22308320, Figure S4B). We now show a similar distribution of the isoforms in the distal colon (see Figure 3): by IB, most of the HNF4α is P2-HNF4α although by staining one can clearly see expression of P1-HNF4α at the top of the crypt (Figure 1).

Even though we are unable to distinguish between the isoforms binding the RELMβ promoter in the Caco2 cells, it is safe to assume that since P2-HNF4α is the major isoform it is certainly binding.

[We also performed parallel ChIP assays with LPS-treated Caco2 cells to recapitulate previously published results showing that LPS can activate the expression of RELMβ in Caco2 cells (PMID:14598255). However, we did not observe any significant, reproducible difference in HNF4α binding with LPS and only a modest, non-statistically significant increase in RELMβ expression, despite using the same range of published LPS concentrations (PMID:14598255). (RELMβ expression was not activated in the HCT116 cells by LPS so we did not attempt ChIP in that cell line.)]

Therefore, our net results (presented in a new Figure 7) are that endogenous HNF4α7 can indeed directly bind the endogenous RELMβ promoter in Caco2 cells in the absence of any additional signals, suggesting that HNF4α can indeed directly activate the expression of the RELMβ gene. Furthermore, we use HNF4 Binding Site Scanner to predict the DNA sequences to which HNF4α binds in the mouse promoter (new Figure 7—figure supplement 1).

Finally, we also attempted HNF4*α* ChIP of mouse colon tissue but due to technical difficulty (sample amount, sonication conditions, etc.) we could not get HNF4α ChIP to work in the allotted time frame (at this point we also had the Caco2 CHIP working so we decided to focus our efforts there). Furthermore, there was good reason to believe that even if we got the CHIP working in the colon, the desired result would not be forthcoming. In Figure 2 we show that in WT mice the level of expression of both isoforms of HNF4α (especially P2-HNF4α) is reduced upon DSS treatment. Therefore, this suggests that since RELMβ expression *increases* with DSS treatment that there is a mechanism other than HNF4α that is involved in the activation of the RELMβ promoter in vivo. We therefore felt that it was important to invest our resources and time to examine that mechanism as well. Those efforts are described below in (2).

2) Indirect regulation of RELMβ expression by HNF4α isoforms

Since we demonstrated differential migration and ion transport in α1HMZ and α7HMZ mice (Figure 5), alterations in RELMβ expression could also be due to other cellular processes altered by the HNF4α isoforms. It is well documented that intestinal permeability is an early event in IBD pathogenesis both in human patients and mouse models. Therefore, RELMβ expression and activation in the α7HMZ mice could be influenced by altered epithelial barrier function. Indeed, it has been shown previously that following DSS treatment there is an increase in intestinal permeability and RELMβ expression (PMID:16815164). These observations led us to hypothesize that altered barrier function could be an alternative mechanism for increased RELMβ expression in and susceptibility of α7HMZ mice to DSS.

To test this hypothesis we conducted several in vivo intestinal permeability assays using FITC-dextran. We found that untreated α7HMZ mice exhibit a moderate (trending towards significance) increased permeability / reduced barrier function compared to WT controls in both untreated and DSS-treated mice (new Figure 7). We also observed that α1HMZ mice have improved barrier function post DSS treatment compared to α7HMZ (6d DSS + 3d recovery) (Figure 7), supporting accelerated recovery as observed in Figure 5.

This new finding prompted us to re-examine our microarray results for dysregulation of genes that could contribute to barrier function: we found several contenders, which are now presented in a new Figure 7. In particular, an increase in *Il4i1* and *Il13ra2* expression in α7HMZ mice could contribute to increased RELMβ expression as reported previously (PMID: 19995957, PMID: 15340149). A decrease in cell adhesion and ion transport genes (Supplementary table S2C and S2D, respectively) could also contribute to increased paracellular permeability and loss of epithelial barrier. Indeed, we have recently shown that overexpression of P2-driven HNF4α (HNF4α8) in HCT116 cells increases cell migration, suggesting a loss of cell adhesion (PMID: 26240283). Loss of E-cadherin (*Cdh1),* a well-established target of HNF4α, aggravates colitis and impairs bacterial defense (PMID:25634675, PMID: 21179475): thus the 4-fold decrease in *Cdh1* expression observed in α7HMZ colons could contribute to impaired barrier function. It is also important to note that *Cdh1* slows epithelial cell migration, as we observe in

α7HMZ mice (PMID: 21179475). Finally, in a paper published just last week in Cell, the authors report that increased IL-18 signaling in intestinal epithelial cell disrupts barrier function (Nowarski et al. Cell, 2015). We found that *Il18r1* and *Il18rap* expression are increased in

α7HMZ mice, while *Il18bp*, a decoy receptor for *Il18* that attenuates downstream signaling is decreased in α7HMZ mice.

Taken togetherwe have now elucidated both a direct and indirect mechanism for the regulation of RELMβ (and hence colitis) mediated by the HNF4α isoforms. In the process we have added another characteristic to the list of phenotypic differences of the HNF4α isoform-specific mice (revised Figure 7).

Reviewer #1:

This paper presents useful data, but it belongs in a more specialized journal. There is already a considerable body of evidence indicating that the HNF4α isoforms are differentially regulated and have different functions. In particular, there is considerable evidence that the P1 form acts as a tumor suppressor. Also, over-expression of HNF4α is oncogenic despite the P1 form, suggesting that P2 acts as an oncogene. The authors nicely test this by generating mice that can only express either the P1 or P2 isoforms and assessing various phenotypes including gene expression profiles and colitis susceptibilities, the authors significantly improve the understanding of the distinct roles of these isoforms. To this non-expert, the experiments generally seem fine with clear results.

We thank the reviewer for finding the manuscript of functional relevance and for insightful comments. We agree with the reviewer that it is now established that P1-HNF4α acts as tumor suppressor in the liver but as we indicate in the Introduction the role of P2-HNF4α in colon cancer is ambiguous with the only work directly addressing the issue being our paper that was published in Oct 2015 in MCB (PMID:26240283). That work uses only HCT116 cells in a xenograft model and it does not explore colitis or colitis-associated colon cancer in an in vivo model as we do here.

There is already a considerable body of evidence indicating that the HNF4α isoforms are differentially regulated and have different functions.

We must respectfully disagree with the reviewer on this point. Having originally cloned HNF4α 25 years ago, we are well aware of the HNF4α literature. Indeed, it has only been recently that a role for HNF4α in cancer and the colon in general has begun to be investigated, as we summarize in the Introduction. The role of the HNF4α isoforms had not been addressed under any pathological conditions in vivo until the development of monoclonal antibodies to the HNF4α isoforms in 2006 (PMID:16400631). In this paper the authors showed for the first time that expression of P1-HNF4α but not P2-HNF4α is lost in colon cancer patients. However, they did not address either the mechanism responsible for the isoform-specific loss of HNF4α nor the functional relevance the isoforms. In fact, it was the observations in this paper that prompted us to investigate the mechanism responsible for the differential loss, which we did in our 2012 PNAS paper (PMID: 22308320) where we showed HNF4α isoform-specific effects of Src tyrosine kinase in human colon cancer.

However, while the PNAS paper confirmed the importance of tracking changes in individual HNF4α isoforms in colon cancer (and provided the mechanism for the isoform-specific down regulation), it did *not* address the function of the isoforms in the colon as we do nowin the current work. In fact, this manuscript is only the second publication on the exon swap (HNF4α isoform-specific) mice, which were created by one of us (NB). The first paper (PMID:16498401) examined the liver of the mice under fed and fasted conditions and proved for the first time that the AB domain of HNF4α has a function in vivo. It did not, however, examine the colon or intestines. Finally, before 2006 there were a couple of papers on the function of the HNF4α isoforms using cell based systems (including one that FMS is co-author on) but as relevant as that work was at the time, it did not examine the isoforms in vivo nor in a global fashion. Indeed, most papers on HNF4α do not even distinguish which isoforms are being examined, which has led to some of the confusion in the field.

In contrast to the paucity of literature on the HNF4α isoforms, despite their importance, there are >1000 publications in Pubmed that come up with “progesterone receptor and isoforms” and yet we still do not understand the difference in the function of the PR isoforms, even though it is well established that they too are dysregulated in cancer. Furthermore, the PR isoforms have not been examined using isoform-specific mouse models as we do here for HNF4α.

Thus, it is a bold overstatement to say that there is a considerable body of evidence on the HNF4α isoforms. Indeed, if we felt that all the questions had been addressed in this regard we would not be spending our time and effort on the topic.

However, the differences between the isoforms are phenomenological with limited mechanistic understanding, and their opposing roles in colitis and colitis-associated colon cancer are in accord with previous characterization as tumor suppressing and tumor promoting. As such, I think this paper will be of interest to specialists in HNF4α and colitis/colon cancer, but it is unclear what findings are of more general significance.

Again, we must respectfully disagree. As described above we did provide a mechanism via RELMβ in our first submission. We have now added two additional mechanisms outlined above in terms of direct binding of HNF4α to the RELMβ promoter as well as alterations in barrier function, along with potential genes that could be responsible for the effects.

Furthermore, colitis is not necessarily the same as colon cancer (although it can lead to it), and hence functional roles of tumor suppressing and tumor promoting may not apply.

Indeed, we examined the effect of chronic DSS treatment in α7HMZ mice as we thought that it would increase tumor load. However, as noted in the text, we found that despite the sensitivity of the α7HMZ mice to colitis and AOM+DSS, DSS treatment alone was not sufficient to induce tumor formation in α7HMZ mice.

Except for a recent paper on the role of P2-HNF4α in gastric cancer (PMID: 25410163), we are not aware of any other literature describing an oncogenic role for P2-HNF4α (Our 2015 MCB paper just shows that P2-HNF4α is permissive of tumor growth, it does not accelerate it.)

*It is already known that loss of RELMβ causes resistance to colitis. The new finding here is this gene is up-regulated by the P2 form, which of course also stimulates colitis. So, the knock-out result is largely expected. Moreover, the conclusion that P2-mediated up-regulation of RELMβ is important is not really tested. The correct experiment would be to knock out HNF4 sites that are required for the up-regulation of RELMβ and show that this blocks colitis. Alternatively, some other manipulation that reduces RELMβ levels to the uninduced level. To put it differently, one has to show that the regulation of RELMβ is important, not just the gene itself which is already known. I realize this is a fair amount of work.*

Prior knowledge that RELMβ contributes to colitis and our finding of increased RELMβ in α7HMZ mice areexactlywhat prompted us to addresswhether this cytokine contributes to increased susceptibility to DSS-induced colitis. Just because RELMβ expression was upregulated, it did not necessarily have to be the case that it was a major player in α7HMZ susceptibility. Fortunately for us, after taking a year to generate the double transgenic mouse and another year to analyze them, upregulation of RELMβ did turn out to be a causal effect.

Please see our response above about the two mechanisms that we have now elucidated for the upregulation of RELMβ in α7HMZ mice – direct regulation by P2-HNF4α as evidenced in a ChIP assay and indirect regulation via altered barrier function.

In the direct mechanism we identify several putative HNF4α binding sites and show that P2- HNF4α can indeed directly bind the RELMβ promoter (new Figure 7 and Figure 7—figure supplement 1). Therefore, knocking out of an HNF4α binding site is not so simple – at least 3 sites would most likely have to be deleted.

Reviewer #2:

HNF4α expression in the gut has been linked to both colitis and colon cancer. Chellapa et al. describe strikingly differential effects of distinct isoforms of HNF4α in the gut. Expression of the P2 promoter isoform only results in increased sensitivity to DSS and increased tumorigenesis in an AOX/DSS model, while opposite results are observed with mice that express only the P1 isoforms. Within the context of nuclear receptor function, it is well known that there are multiple isoforms of many NRs, but they are generally not distinguished functionally. Thus, this represents a particularly clear example of delineation of distinct functions of different isoforms. This observation might be of more limited interest, but clear mechanistic information is provided by the demonstration that RELMβ is overexpressed in the DSS sensitive P2-specific mice, and that doubly mutant P2 specific/RELMβ knockout mice are resistant to the increased sensitivity to DSS.

We thank the reviewer for his in-depth understanding of our manuscript and its relevance to both the nuclear receptor field and colon pathology.

One question raised by these results is whether the effects of the different isoforms on RELMβ expression are direct or indirect, and particularly whether the distinct isoforms have distinct direct effects on RELMβ expression. This study would be further strengthened if relatively straightforward RELMβ promoter cotransfections showed such effects, or if other strategies could document differential promoter occupancy.

In response to the reviewers’ comments, as described above, we performed a ChIP assay and found that endogenous HNF4α in a human colon epithelial cell line (Caco2) can bind the endogenous RELMβ promoter directly. We also found that there are predicted HNF4α binding sites in the same region of mouse RELMβ gene (see new Figure 7, Figure 7—figure supplement 1). This makes RELMβ a direct target of HNF4α.

We do not feel that co-transfection assays would be any more informative than the expression profiling (and now ChIP). In our experience in working with HNF4α for the last 25 years, putting a promoter with an HNF4α binding site in front of a luciferase gene and co-transfecting in HNF4α will result in activation of the promoter, typically in direct relation to the affinity that HNF4α has to the site in vitro.

At this point, as mentioned abovein the overall response to the editor, we are unable to determine whether P2-HNF4α but not P1-HNF4α can bind the RELMβ promoter in vivo. However, also as described above, we predict that P1-HNF4α as well as P2-HNF4α would be able to bind the RELMβ promoter based on our recently published work in MCB (PMID: 26240283) and that different co-regulators and transcription factors recruited to the promoter confer the specificity. We do find the issue of differential regulation despite similar DNA binding to be a very interesting and important one. We are trying to pursue it in the liver where there is much more tissue available but it is not a trivial issue and definitely beyond the scope of the current work.

Complicating the situation is the issue of which isoforms are present during DSS treatment when RELMβ expression is elevated. In Figure 2 we show that both P1- and P2-HNF4α are down regulated at 6 days DSS (which would make ChIP even more difficult if not impossible). In Figure 4 we also show that in α7HMZ mice there is an upregulation of P2-HNF4α at 6 days DSS. While the reason for this increase remains unknown it could nonetheless explain the upregulation of RELMβ at that time point in those mice compared to WT mice – there is simply more of P2-HNF4α present. It also suggests that other factors are at play in regulating RELMβ expression.

As described above, we also now elucidate an indirect, as well as direct, mechanism of regulation of RELMβ. We show using FITC-dextran studies that the α7HMZ mice have a lower barrier function than WT mice, which could allow more bacteria to cross the barrier and hence activate expression of the RELMβ gene.

In summary, we believe that, there could be two different mechanisms occurring. In the untreated mice the lower basal level of barrier function in α7HMZ mice could lead to increased signaling from bacteria that cross the barrier and hence activation by CDX2 (PMID:15576623) even though there is no difference in the amount of HNF4α that is present compared to WT mice (see Figure 3). While in the DSS-treated mice, in addition to even lower barrier function and hence more bacteria presumably crossing the barrier, in the α7HMZ mice there is more HNF4α present than in the WT mice which could also contribute to greater RELMβ expression.

[Editors’ note: the author responses to the re-review follow.]

The two re-reviews are very similar. While the new experiments improve the manuscript, they don't address the key question that was required, namely no mechanistic connection between the 2 isoforms, expression of RELMβ, and the phenotype. Perhaps the authors could perform an experiment (one suggested below) that would address this, but at this point the paper is not acceptable. If the authors do such an experiment, it would be reasonable to reconsider on appeal.

*Reviewer #1:*

*The authors have provided thoughtful responses to my comments, and they have done some new experiments that are useful. Nevertheless, I don't think they fully understood my point of view, and they only minimally addressed the key issue.*

1) The paper clearly represents an advance over published work, the experiments are well done, and it deserves publication. I don't dispute any of the authors' responses about how their paper is advanced over previous work. However, the relevant question is how significant the advance is over previous work, particularly at the "general interest" level needed for publication in eLife. From my outside perspective, there is already evidence (perhaps considerable was too strong) that P1 acts as a tumor suppressor and that P2 acts an oncogene (less explicitly demonstrated, but the only reasonable way to explain how overproduction of HNF4α is oncogenic). The authors experiments are a nice and more explicit demonstration of the opposing roles of the P1 vs P2 forms, but I do not view this is as a major advance. Yes, the experimental systems are different and the experiments better, but the basic point is the same. For this reason, I (and the other Reviewer) thought that the basic opposing phenotypes were worth publishing, but not a sufficient advance for eLife.

We thank the reviewer for acknowledging the importance of the work but for the record we respectfully disagree that we had done nothing more than prove our point with more convincing and relevant systems. In addition to showing differential in vivo effects of P1-HNF4α and P2-HNF4α on colon cancer, we also showed that they play different roles in colitis as well as barrier function, RELMβ expression and different distribution within the tissue (none of this had been shown previously).

While perhaps the reviewer does not consider that these results with clinical implications are of sufficient importance for *eLife*, we would argue that showing how disruption of the balance of splice variants in vivo can result in disease is an appropriate topic for *eLife*, especially considering that >50% of human genes have alternative promoters and >90% of human genes undergo alternative splicing even though for most genes we know next to nothing about the impact of those splice variants.

In addition to parsing the in vivo role of the HNF4α isoforms in the colon, colitis and colon cancer, to our knowledge this work is also the first to document a change in promoter usage/transcript variant along the colonic crypt that results in an induction of colitis and/or colon cancer. Changes in expression of different genes along the crypt are well documented (e.g., APC/b-catenin, Ki67, NKCC1) but on 4/30/16 there are only 10 hits in Pubmed in a search for “(colonic crypt) AND alternative splicing,” including some where alternative splice variants are associated with colon cancer (e.g., CD44) but none that show the change in variants actually *induce* colon cancer or colitis susceptibility, as we show here for HNF4α.

A search of Pubmed for “exon swap” yields no papers published on this topic in *eLife*. There are only 4 papers published in *eLife* on “alternative promoter” (none discussing the generation of different transcript variants from those promoters) and only 21 out of 2360 total papers in *eLife* on “alternative splicing”. These numbers indicate that as important as alternative splicing and alternative promoter usage are, they remain highly understudied.

2) Because of point 1, I viewed it as important to have some mechanistic understanding of how the distinct HNF4α isoforms are linked to RELMβ. Of course, the primary finding (point 1) means that mice expressing these different isoforms have very different phenotypes and hence very different transcriptional profiles. Ultimately, this means that the different isoforms have different transcriptional effects, but profiling data per se provides no mechanistic information. Conversely, the previous knowledge that RELMβ is linked to colitis combined with the authors' finding that RELMβ levels are increased in P2 mice does not indicate that this up-regulation is functionally important. It is entirely possible that a RELMβ derivative expressed at its normal level but unable to be induced by P2 would behave similarly to the wt (inducible) allele. To put it differently, one predicts that increased DSS sensitivity would depend on RELMβ no matter how once increased DSS sensitivity is achieved. So, again, there is no specific link between P2-overexpression and the RELMβ requirement for DSS sensitivity.

We now show that the HNF4α isoforms differentially activate the RELMβ promoter in a revised Figure 7 using the previously suggested luciferase assays. The results show that P2-HNF4α does indeed activate the RELMβ promoter more potently than P1-HNF4α. We humbly thank the Reviewer for insisting that we perform this experiment. (Please see specifics below.)

To address the next issue regarding the mechanism by which the HNF4α isoforms activate transcription in a differential fashion, we performed a RIME assay in which we identify in a global, unbiased fashion the proteins that the isoforms interact with in the colon. As we predicted from our unpublished work in the liver, we found >100 differentially interacting proteins which we now present in a revised Figure 5. Any one of these differentially interacting proteins could have an impact on HNF4α transactivation. For example, we found that Src tyrosine kinase specifically interacts with P1-HNF4α but not P2-HNF4α in mouse colon. This finding validates our approach as it is consistent with our 2012 PNAS work in which we show that in cell-based assays Src specifically phosphorylates P1-HNF4α and causes a decrease in protein stability as well as nuclear localization and transactivation potential. In this single experiment in Figure 5 we now add to the Src story more than a dozen other kinases, phosphatases and other proteins that could impact HNF4α transactivation.

We respectfully disagree that there is no specific link between “P2-overexpression” (expression in different compartment of the crypt would be more accurate) and the RELMβ requirement for DSS sensitivity. It is not clear to us how the double transgenic RELMβ KO and α7HMZ exon swap mice that lose the remarkable lethality to DSS does not prove the necessity of RELMβ. It is true that α7HMZ may promote colitis through multiple pathways, but these are evidently all ultimately *dependent* on RELMβ, since deleting RELMβ abrogates α7HMZ-induced lethality.

It is not clear what is meant by the following, but we will address it to the best of our ability:

“It is entirely possible that a RELMβ derivative expressed at its normal level but unable to be induced by P2 would behave similarly to the wt (inducible) allele. To put it differently, one predicts that increased DSS sensitivity would depend on RELMβ no matter how once increased DSS sensitivity is achieved.”

The KO removed all exons in the *Retnlb* gene. We realize that many different factors, in addition to HNF4α, affect RELMβ expression. Indeed, that is why we took the approach of looking at the effect of the HNF4α isoforms on barrier function as that would be a starting point of multiple pathways to RELMβ expression.

We have now added to the barrier function pathway, the pathway of a7HMZ directly activating the RELMβ promoter (see below re revised Figure 7).

3) The new ChIP experiments are useful, because they show that, in principle, the HNF4α isoforms could directly affect transcription of RELMβ. However, this does not really address the critical point, namely the differential behavior of the isoforms. Also, the focus on DNA-binding is possibly misplaced, as it is entirely possible that the distinction between the isoforms relates to transcriptional potential or interaction with other proteins (with differential transcriptional potential). I agree that artificial reporters with HNF4α sites is likely to be useless. But, what I imagined was an experiment with a reporter driven by a reasonably large segment of the RELMβ locus and co-transfection of the individual isoforms.

To address the transactivation potential of HNF4α isoforms on RELMβ promoter we performed luciferase assays on RELMβ reporter constructs containing up to 800 bp of promoter sequence as well as HNF4α binding sites (Figure 7—figure supplement 1) that we identified recently in silico and in ChIP assay (Figure 7). We co-transfected HNF4α isoforms in LS174T cell line with RELMβ reporter constructs and we found that P2-HNF4α activates RELMβ promoter significantly better than P1-HNF4α (new Figure 7).

Furthermore, we also knocked down the endogenous expression of HNF4α isoforms in LS174T cell line and found that knock down of P2-HNF4α significantly decreases RELMβ promoter activity in the luciferase assay compared to siP1-HNF4α (new Figure 7—figure supplement 1). These experiments show that both HNF4α isoforms regulate RELMβ promoter but P2-HNF4α is stronger transactivator than P1-HNF4α. On the other hand P2-HNF4α does not activate ApoB promoter well compared to P1-HNF4α (new Figure 7—figure supplement 1), indicating that there are promoter-specific effects of the HNF4α isoforms, consistent with the gene expression arrays. The findings are also consistent with those in the revised Figure 5 where we show that the HNF4α isoforms interact with different proteins, which could result in differential transactivation potential.

4) The new barrier function experiments are nice on their own, but don't address mechanism related to the distinct isoforms. Again, the P1 vs. P2 cells have many differences, but those differences per se don't address any mechanistic understanding. The effects on barrier function and/or RELMβ could be very indirect consequences of the distinct HNF4α isoforms. By definition, the phenotypic differences caused by the distinct isoforms means that, at some level any property (molecular or overall phenotype) is differentially affected. In addition, the relationship of the P2 mice vs. other versions of DSS-induced colitis means that there are mechanistic similarities between those situations, but they don't address be basis of why P1 vs. P2 are different.

RIME analysis in a new Figure 5 addresses this issue as well as it can be addressed given the current technology. The only thing that would be left to do is determine which of the 100+ differentially interacting proteins is relevant for each of the 100+ HNF4α target genes. This is clearly beyond the scope of any single paper, regardless of the journal.

For these reasons, although the manuscript is improved, I still believe that the differences between the isoforms are phenomenological with limited mechanistic understanding, and their opposing roles in colitis and colitis-associated colon cancer are in accord with previous characterization as tumor suppressing and tumor promoting. As such, I think this paper will be of interest to specialists in HNF4α and colitis/colon cancer, and certainly worth publishing in a specialized journal, but it is unclear what findings are of more general significance.

We thank the reviewer for acknowledging improvements to the manuscript and are very appreciative of his/her time and effort in reviewing it. We sincerely hope that we have now provided sufficient evidence to show that the HNF4α isoforms activate RELMβ and other genes in a differential fashion due to unique sets of interacting proteins, and that we have convinced him/her that our manuscript is worthy of publication in *eLife*.

Reviewer #2:

This revised manuscript has been improved by the direct demonstration that RELMβ is an HNF4α target gene, and also the identification of the differential barrier function in the two isoform specific strains. However, the primary requirement for acceptance of this manuscript was clearly stated in the reviewing editor's decision: "What is missing is a mechanistic connection between the 2 isoforms and expression of RELMβ. Thus, publication in eLife requires making such a mechanistic connection…" As the authors acknowledge, this is a difficult, but important issue. Thus, despite the improvements, this revised manuscript has not effectively answered the primary question.

We have addressed the direct regulation of RELMβ by HNF4α isoforms using a luciferase reporter assay, as requested, and show that P2-HNF4α is a stronger transcriptional activator of RELMβ promoter compared to P1-HNF4α in a revised Figure 7 as well as a new panel in the supplement to Figure 7.

We also addressed the issue of the mechanism by which the HNF4α isoforms might differentially regulate gene expression by showing in a RIME (aka proteomics) assay in a revised Figure 5 that the HNF4α isoforms interact with different proteins in vivo, particularly signaling factors and RNA binding proteins that could alter transactivation activity.

From our Response to Reviewer #1:

To address the next issue of regarding the mechanism by which the HNF4α isoforms activate transcription in a differential fashion, we performed a RIME assay in which we look in vivo at what proteins the isoforms are interacting with. As we predicted from our unpublished work in the liver, we found >100 differentially interacting proteins which we now present in a revised Figure 5. Any one of these differentially interacting proteins could have an impact on HNF4α transactivation. For example, we found that Src tyrosine kinase specifically interacts with P1-HNF4α but not P2-HNF4α in mouse colon. This finding validates our approach as it is consistent with our 2012 PNAS work in which we show that in cell-based assays Src specifically phosphorylates P1-HNF4α and causes a decrease in protein stability as well as nuclear localization and transactivation potential. In this single experiment in Figure 5 we now add to the Src story more than a dozen other kinases, phosphatases and other proteins that could impact HNF4α transactivation.